# Effective Adsorption of Methylene Blue dye onto Magnetic Nanocomposites. Modeling and Reuse Studies

**Silvia Alvarez-Torrellas [1],\*, Mokhtar Boutahala [2],\*, Nadia Boukhalfa [2] and Macarena Munoz [3]**

[1] Catalysis and Separation Processes Group (CyPS), Chemical Engineering and Materials Department, Faculty of Chemistry Sciences, Complutense University, Avda. Complutense s/n, 28040 Madrid, Spain; nadouchette2011@hotmail.fr

[2] Laboratory of Chemical Process Engineering, Department of Process Engineering, Faculty of Technology, University of Ferhat Abbas Setif-1, Setif 19000, Algeria

[3] Chemical Engineering Department, Faculty of Sciences, Autónoma University, Cantoblanco, Ctra. Colmenar km 15, 28049 Madrid, Spain; macarena.munnoz@uam.es

\* Correspondence: satorrellas@ucm.es (S.A.-T.); mboutahala@yahoo.fr (M.B.); Tel.: +34-91-394-4118 (S.A.-T.); +213-(0)-699-13-63-94 (M.B.)

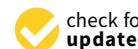

**Featured Application: Treatment of polluted water with dyes in textile and paper industries.**

**Abstract:** In the present study, new adsorbent beads of alginate (A)/maghemite nanoparticles ($\gamma$-Fe$_2$O$_3$)/functionalized multiwalled carbon nanotubes (f-CNT) were prepared and characterized by several techniques, e.g., N$_2$ adsorption-desorption isotherms, Fourier transformed infrared spectroscopy (FTIR), X-ray diffraction (XRD), thermogravimetric analysis (TGA/DTG), scanning electron microscopy (SEM), and vibrating sample magnetometry (VSM) and further tested for the adsorption of the dye methylene blue (MB) from water. The beads (A/$\gamma$-Fe$_2$O$_3$/f-CNT) presented a relatively low BET specific surface area value of 59 m$^2$g$^{-1}$. The magnetization saturation values of A/$\gamma$-Fe$_2$O$_3$/f-CNT beads determined at 295 K was equal to 27.16 emu g$^{-1}$, indicating a magnetic character. The time needed to attain the equilibrium of MB adsorption onto the beads was estimated within 48 h. Thus, several kinetic and isotherm equation models were used to fit the kinetic and equilibrium experimental results. The number of adsorbed MB molecules per active site, the anchorage number, the receptor sites density, the adsorbed quantity at saturation, the concentration at half saturation and the molar adsorption energy were quantified using the monolayer model. The calculated negative $\Delta G^0$ and positive $\Delta H^0$ values suggested the spontaneous and endothermic nature of the adsorption process. In addition, A/$\gamma$-Fe$_2$O$_3$/f-CNT composites can be used at least for six times maintaining their significant adsorptive performance and could be easily separated by using a magnet from water after treatment.

**Keywords:** adsorption; carbon nanotubes; composite; maghemite; methylene blue

## 1. Introduction

Dyes are widely used in many industrial sectors such as textile and paper companies [1]. These industries dump large amounts of wastewater into streams without any prior treatment. Their discharges present a source of pollution that can be very harmful to human and environmental health. The protection of the environment and the improvement of water quality by means of the treatment of these industrial effluents have become an issue of concern in recent years. Methylene blue (MB) is a synthetic dye considered a suspected carcinogenic agent. It can cause breathing difficulties, nausea

and vomiting [2,3], as well as allergic dermatitis and skin irritation [4], so the efficient removal of this contaminant from water is of high scientific relevance.

Several technologies to remove these dyes from wastewater have been reported, such as adsorption and membrane separation [5–7]. Among these techniques, activated carbon adsorption is an effective method for treating contaminated wastewater, widely supported in the literature. Conventional activated carbon materials are commonly used as adsorbents; however, the difficulties for the separation after treatment in some applications limit their use and, in some cases, make the process unfeasible for industrial applications [7,8].

As it has been widely reported in the literature, conventional activated carbons, synthesized from a single precursor, are economical materials used in the adsorbent processes, and even finding specific applications, such as supercapacitors [9] and catalysts [10], due to their relatively low cost.

Recently, nanomaterials have been intensively used as adsorbents to remove organic substances from wastewater, showing great potential in the environmental remediation field due to their moderate-large specific surface area and opened-porous structure [11–13]. Among these nanomaterials, carbon nanotubes (CNTs) [14,15] and magnetic iron oxides [16,17] are considered as excellent adsorbents for dyes removal.

CNTs can be classified attending to two categories, depending on their size and structure: single-walled carbon nanotubes (SWCNTs, 0.7 < d < 2 nm) and multi-walled carbon nanotubes (MWCNTs, 1.4 < d < 150 nm) [13]. MWCNTs have showed excellent properties for the adsorption of dyes, pesticides, heavy metal ions and other organic pollutants from water. They possess small radial size, excellent mechanical properties and exceptional electrical and thermal conductivity [18]. Chemical modifications such as chemical oxidation of MWCNTs lead to incorporate new functional groups on their surface, enhancing their affinity and selectivity towards the target pollutants [14]. In addition, these functionalities can increase the MWCNTs solubility, which help to increase the interactions between MWCNTs surface and the contaminant [18]. However, the use of MWCNTs as nano-adsorbents is limited due to their separation and regeneration difficulties.

The most used iron oxides in different technological applications are hematite ($\alpha$-$Fe_2O_3$), magnetite ($Fe_3O_4$) and maghemite ($\gamma$-$Fe_2O_3$) [17]. These oxide materials demonstrated a huge potential for their application in wastewater treatment due to their, magnetic properties, relevant selectivity, reactivity and biocompatibility abilities [19]. They have also been used in different research areas including catalysis, environmental remediation, magnetic storage media, biomedicine, ferrofluid, magnetic resonance imaging and electrochemical applications [20]. However, the disadvantages of these materials, in particular magnetite and maghemite nanoparticles (MNPs), is that they tend to aggregate between them due to magnetic dipole-dipole attraction forces. In addition, MNPs are prone to oxidation [21], corrosion by acidic or basic medium [22] and release of metal ions [23]. Moreover, it has been shown that the improvement of the adsorption capacity and the chemical stability of magnetic nanoparticles can be enhanced by their surface stabilization by incorporating functional groups such as amine, carboxylic and phosphoric acids, or its coverage by dispersing into activated carbon, graphene, and carbon nanotubes to form composite materials [8,11,16,17,24–26].

Sodium alginate (A) is a linear biopolymer. It has been extracted from brown algae and consists of edible polysaccharides with homopolymeric blocks of 1–4 linked d-mannuronated and L-guluronate. As adsorbent, sodium alginate is effective for the removal of several heavy metal ions. Its adsorptive ability comes from its structure, which presents many carboxylic functional groups [27]. This biopolymer is environmentally benign, biodegradable, hydrophilic and abundant in nature [5,7]. Because of these advantages, alginate has been usually used as beads to encapsulate several adsorbents, including magnetite nanoparticles [28], activated carbon [29], carbon nanotubes [30], graphene oxide [31], titania nanoparticles [32] and clays [33] in order to improve their dispersion, stability and affinity towards several compounds in aqueous solution.

It is known that a certain amount of the materials used in the adsorption processes could be lost when they are used in powder form after water treatment, generating another environmental problem

and huge quantities of waste. Likewise, these powdered materials cannot be used in a fixed-bed column because of the regeneration costs and the clogging of the reactor [34]. In general, the encapsulation of the adsorbent in alginate beads is an excellent solution to overcome this problem. In this context, the purpose of this study is to prepare an adsorbent material by encapsulating functionalized maghemite $\gamma$-Fe$_2$O$_3$ and carbon nanotube (f-CNT) in alginate beads (A) and used them for methylene blue uptake.

## 2. Materials and Methods

### 2.1. Materials

Methylene blue, sodium alginate, FeCl$_2$ 4H$_2$O (98%) and FeCl$_3$·6H$_2$O (45%) solutions were purchased from Sigma-Aldrich (Darmstadt, Germany). Commercial acid functionalized multiwalled carbon nanotubes were supplied from a local supplier in Malaysia (Johor Bahru) and were prepared by chemical vapour deposition. A schematic figure of the tested multiwalled carbon nanotubes is depicted in the Supplementary Material (Scheme S1). The dimensions provided in the scheme have been estimated by TEM analysis, considering opened carbon nanotubes. Fe(NO$_3$)$_3$ solution (99%) was obtained from Emory Healthcare (Bahasa, Malaysia) and sodium citrate (99%) was supplied by Bio Basic Inc. (Puchong, Malaysia). Calcium chloride (99%) was purchased from Fluka (Reinach, Switzerland). Nitric acid solution (65%), acetone (99.8%), diethyl ether (99.9%) and ammonia solution (25%) were obtained from QReC (Rawang, Malaysia). Finally, ethanol (Normapur 99.96%) was supplied by AnalaR (Pelatilg Jaya, Malaysia).

### 2.2. Preparation of Magnetic $\gamma$-Fe$_2$O$_3$ Nanoparticles

$\gamma$-Fe$_2$O$_3$ nanoparticles synthesis was carried out following a modified procedure described by Bee et al. [28]. Thus, magnetite particles were prepared by co-precipitation after adding ammonium hydroxide solution to the mixture of FeCl$_2$ 4H$_2$O and FeCl$_3$ 6H$_2$O solutions. Then, magnetite particles were oxidized to maghemite in nitric acid solution at 90°C using a boiling Fe(NO$_3$)$_3$ solution. Maghemite nanoparticles were then isolated and rinsed using acetone and ether and dispersed in water. In order to create a negative charge and to stabilize the surface of the obtained nanoparticles, maghemite was coated by adding 5 g of sodium citrate to 50 mL of magnetic solution at 90 °C for 90 min. Citrate functionalized maghemite nanoparticles were dispersed in water, affording a brown ferrofluid with a pH value of 7.7.

### 2.3. Preparation of (A/$\gamma$-Fe$_2$O$_3$/f-CNT) Beads

The composite beads (A/$\gamma$-Fe$_2$O$_3$/f-CNT) were synthesized as follows: 2 g of sodium alginate was dissolved in 100 mL of dispersion containing 16 mL of citrate functionalized ferrofluid. 0.4 g of f-CNT was then added to the solution at continuous stirring. This magnetic solution was dropped in a calcium chloride solution (4%) in order to form the beads. The beads were gently stirred for 24 h in calcium solution to obtain the total gelation of the alginate. After 24 h, A/$\gamma$-Fe$_2$O$_3$/f-CNT beads were separated, washed and stored in distilled water. Magnetic beads without f-CNT were also prepared by the same method and named as A/$\gamma$-Fe$_2$O$_3$.

### 2.4. Characterization Techniques

The textural properties of MWCNTs, $\gamma$-Fe$_2$O$_3$ and A/$\gamma$-Fe$_2$O$_3$/f-CNT were explored by adsorption-desorption of N$_2$ at 77 K in an ASAP 2020 system (Micromeritics, Norcross, GA, USA). Before the analysis, the samples were outgassed at 250 °C for 3 h. The specific surface area ($S_{BET}$, m$^2$ g$^{-1}$) of the materials was evaluated by Brunauer, Emmett and Teller (B.E.T.) equation in a pressure range of P/P$^0$ = 0.15–0.35. In addition, the external surface area ($S_{ext}$, m$^2$ g$^{-1}$) was determined according to *t-plot* method and the total pore volume value ($V_{Total}$, cm$^3$ g$^{-1}$) considered was at a relative pressure of P/P$^0$ = 0.99. FTIR spectra of $\gamma$-Fe$_2$O$_3$, f-CNT, A, A/$\gamma$-Fe$_2$O$_3$ and A/$\gamma$-Fe$_2$O$_3$/f-CNT before and after MB adsorption were obtained in a FTIR 8400S spectrometer (Shimadzu, Kyoto, Japan) in the range of 400-4000 cm$^{-1}$,

using KBr pellet technique. The magnetic properties of $\gamma$-Fe$_2$O$_3$, A/$\gamma$-Fe$_2$O$_3$ and A/$\gamma$-Fe$_2$O$_3$/f-CNT were determined using a 7404-S vibrating sample magnetometer (Lake Shore, Princeton, Wi, USA) at room temperature. The crystalline phases in $\gamma$-Fe$_2$O$_3$, f-CNT and A/$\gamma$-Fe$_2$O$_3$/f-CNT materials were analysed by X-ray diffraction (XRD) using an X´Pert PRO diffractometer (Malvern Panalytical Inc., Westborough, MA, USA) with Cu K$\alpha$ radiation (45kV, 40 mA) in the 2$\theta$ range between 10° and 80° at a scanning rate of 2° min$^{-1}$. The spectra were analyzed by the Rietveld Refinement method using the Material Analysis Using Diffraction (MAUD) program and the average crystal size was calculated using the Debye-Scherrer equation. Thermogravimetric analysis (TGA/DTG) was conducted in an EXTAR 6000 thermal analyzer (Seiko, Mahwa, NJ, USA) under nitrogen atmosphere at a heating rate of 10 °C min$^{-1}$, from 30 to 900 °C. Thus, TGA/DTG analysis in air atmosphere (from 30 to 900 °C, 10 °C min$^{-1}$) of $\gamma$-Fe$_2$O$_3$ particles, f-CNT, alginate/$\gamma$-Fe$_2$O$_3$ beads and A/$\gamma$-Fe$_2$O$_3$/f-CNT composite were accomplished. Finally, the surface morphology of the materials was studied by scanning electron microscopy (SEM) using a JSM 6400 microscope (JEOL, Peabody, MA, USA) equipped with a thermoionic cathode and a 25 kV detector. EDX analysis of A/$\gamma$-Fe$_2$O$_3$/f-CNT was accomplished. Additionally, the morphology of functionalized carbon nanotubes (f-CNT) was explored by transmission electronic microscopy (TEM) in a JEOL JEM 2100 microscope (200 kV, 0.25 nm of resolution).

## 2.5. Adsorption Experiments

The effect of pH on MB adsorption onto A/$\gamma$-Fe$_2$O$_3$/f-CNT was studied varying the initial pH values from 3 to 10. The volume used was of 50 mL, operating at room temperature, ±25 °C. Kinetic batch adsorption tests were accomplished using a MB initial concentration of 230 mg L$^{-1}$. For this, 50 mg of A/$\gamma$-Fe$_2$O$_3$/f-CNT was dispersed in MB solution (50 mL) at natural pH (5.2) and room temperature. At regular time intervals, a sample was magnetic separated. The equilibrium adsorption experiments were accomplished at different initial MB concentrations, ranging from 5 to 270 mg L$^{-1}$, mixed with 50 mg of $\gamma$-Fe$_2$O$_3$/f-CNT. The dispersions were maintained at a constant temperature of 25°C, at a natural pH of 5.2. All the adsorption experiments were carried out without stirring. When the equilibrium was reached, the magnetic adsorbent could be separated from the aqueous medium by using a magnet. The effect of temperature on the adsorption was studied at different temperatures, i.e., 283, 293, 303 and 313 K using a thermostatic chamber.

MB concentration was determined on a Shimadzu UV-1700 spectrophotometer at a wavelength of 664 nm. The amount of adsorbed dye at equilibrium time (q$_e$, mg g$^{-1}$) was determined using Equation (1):

$$q_e = \frac{(C_0 - C_e) \cdot V}{m} \qquad (1)$$

where, $C_0$ and Ce (mg L$^{-1}$) are MB concentration at initial and equilibrium time, respectively, m (g) is the weight of adsorbent and V (L) is the solution volume.

## 2.6. Desorption of MB Dye and Reusability Studies

A/$\gamma$-Fe$_2$O$_3$/f-CNT composite beads were used for six consecutive adsorption-desorption cycles in order to investigate their regeneration efficiency and examine their cost effectiveness. After adsorption, the desorption of MB from the beads was carried out using 10 mL of 0.1M HNO$_3$ solution, using 10 mg of adsorbent. The desorption efficiency (DE) was calculated by using Equation (2):

$$DE\ (\%) = \frac{C \cdot V}{q \cdot m} \cdot 100 \qquad (2)$$

where, q (mg g$^{-1}$) is the adsorbed amount of MB, C (mg L$^{-1}$) is the MB concentration in the desorption solution, V (L) is the desorption volume and m (g) is the mass of adsorbent used in the desorption tests.

*2.7. Analysis of the Experimental Data*

In order to find the best correlation of the experimental data, the regression coefficient ($R^2$) and residual root-mean squared error (RMSE) values (Equations (3) and (4), respectively) were calculated using Origin 8.5 software (OriginLab Corporation, Northampton, MA, USA).

$$R^2 = 1 - \frac{\sum\limits_{n-1}^{n} (q_{e,t.\exp.n} - q_{e,t.cal.n})^2}{\sum\limits_{n-1}^{n} (q_{e,t.\exp.n} - \overline{q_{e,t.\exp.n}})^2} \tag{3}$$

$$\text{RMSE} = \sqrt{\frac{1}{n-1}\sum\limits_{n-1}^{n} (q_{e,t.\exp.n} - q_{e,t.cal.n})^2} \tag{4}$$

where, n denotes the number of experimental data, $q_{e,t.\exp}$ and $q_{e,t.cal}$ are the experimental and calculated adsorption capacities at equilibrium or at any time, respectively.

## 3. Results and Discussion

*3.1. Characterization of the Tested Adsorbents*

$N_2$ adsorption-desorption isotherms obtained at 77 K of f-CNT, $\gamma$-$Fe_2O_3$ particles and A/$\gamma$-$Fe_2O_3$/f-CNT beads are depicted in Figure 1. The textural parameters, specific surface area ($S_{BET}$), external surface area ($S_{ext}$), pore volume ($V_{total}$) and pore diameter of the tested materials are collected in Table 1. According to IUPAC classification [35], carbon nanotubes (f-CNT) are materials that show a mainly mesoporous structure; with a contribution of microporosity (~40 $cm^3$ $g^{-1}$ at low P/P$^0$ values) and an important quantity of meso-macropores (type IVa isotherm, according IUPAC classification). These adsorbents show that both brands of $N_2$ adsorption isotherm are parallel and near to the verticality, characteristic of materials with cylindrical pores, where capillary condensation occurs (Table 1). Nevertheless, $\gamma$-$Fe_2O_3$ and A/$\gamma$-$Fe_2O_3$/f-CNT beads showed $N_2$ adsorption isotherms where, at low relative pressure values, the amount of $N_2$ adsorbed is very low (lower than 10 $cm^3$ $g^{-1}_{STP}$). These are isotherms representative of essentially mesoporous material, without poor contribution of microporosity on their structure. Thus, f-CNT material showed a low-moderate specific surface area (265 $m^2$ $g^{-1}$), since relatively low BET area values were found for $\gamma$-$Fe_2O_3$ nanoparticles and A/$\gamma$-$Fe_2O_3$/f-CNT composites, 35 and 59 $m^2$ $g^{-1}$, respectively.

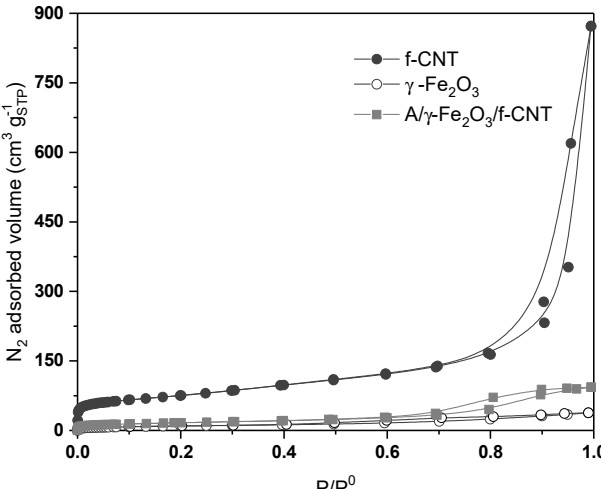

**Figure 1.** $N_2$ adsorption-desorption isotherms at 77 K of f-CNT, $\gamma$-$Fe_2O_3$ particles and A/$\gamma$-$Fe_2O_3$/f-CNT beads.

**Table 1.** Textural parameters of f-CNT, $\gamma$-Fe$_2$O$_3$ and A/$\gamma$-Fe$_2$O$_3$/f-CNT beads.

|  | S$_{BET}$ (m$^2$ g$^{-1}$) | S$_{ext}$ (m$^2$ g$^{-1}$) | V$_{Total}$ (cm$^3$ g$^{-1}$) | Pore Diameter (nm) |
|---|---|---|---|---|
| **f-CNT** | 265 | 331 | 1.35 | 2.5–3.5 [a] |
| **$\gamma$-Fe$_2$O$_3$** | 35 | 31 | 0.06 | 1.0–1.6 [b] |
| **A/$\gamma$-Fe$_2$O$_3$/f-CNT** | 59 | 103 | 0.14 | 1.0–1.5 [b] |

[a] Estimated by BJH model; [b] Estimated by MP method.

The FTIR spectra of (A), f-CNT, $\gamma$-Fe$_2$O$_3$ nanoparticles, A/$\gamma$-Fe$_2$O$_3$ and A/$\gamma$-Fe$_2$O$_3$/f-CNT beads are shown in Figure 2a. A wide band at 3400–3441 cm$^{-1}$ was observed in all spectra and attributed to the stretching vibration of -OH groups coming from the physically adsorbed water. The peak observed at 2923 cm$^{-1}$ for A, A/$\gamma$-Fe$_2$O$_3$ and A/$\gamma$-Fe$_2$O$_3$/f-CNT spectra is due to the antisymmetric stretching vibration of the C-H groups. The absorption band in the range from 1625 to 1616 cm$^{-1}$ found in A, $\gamma$-Fe$_2$O$_3$, A/$\gamma$-Fe$_2$O$_3$ and A/$\gamma$-Fe$_2$O$_3$/f-CNT materials was attributed to the symmetric vibrations of the COO$^-$ groups. The peaks at 1596 cm$^{-1}$ ($\gamma$-Fe$_2$O$_3$), 1415 cm$^{-1}$ (A/$\gamma$-Fe$_2$O$_3$) and 1421 cm$^{-1}$ (A, A/$\gamma$-Fe$_2$O$_3$/f-CNT) were due to the antisymmetric vibrations of COO$^-$ groups [36]. The stretching vibration of C-O groups in the alginate polysaccharide was confirmed at 1052 cm$^{-1}$, 1026 cm$^{-1}$ and 1020 cm$^{-1}$ for A, A/$\gamma$-Fe$_2$O$_3$ and A/$\gamma$-Fe$_2$O$_3$/f-CNT, respectively. Finally the characteristic Fe-O elongation vibration was observed at 622 cm$^{-1}$ in $\gamma$-Fe$_2$O$_3$ and A/$\gamma$-Fe$_2$O$_3$ spectra and at 618 in A/$\gamma$-Fe$_2$O$_3$/f-CNT [37,38].

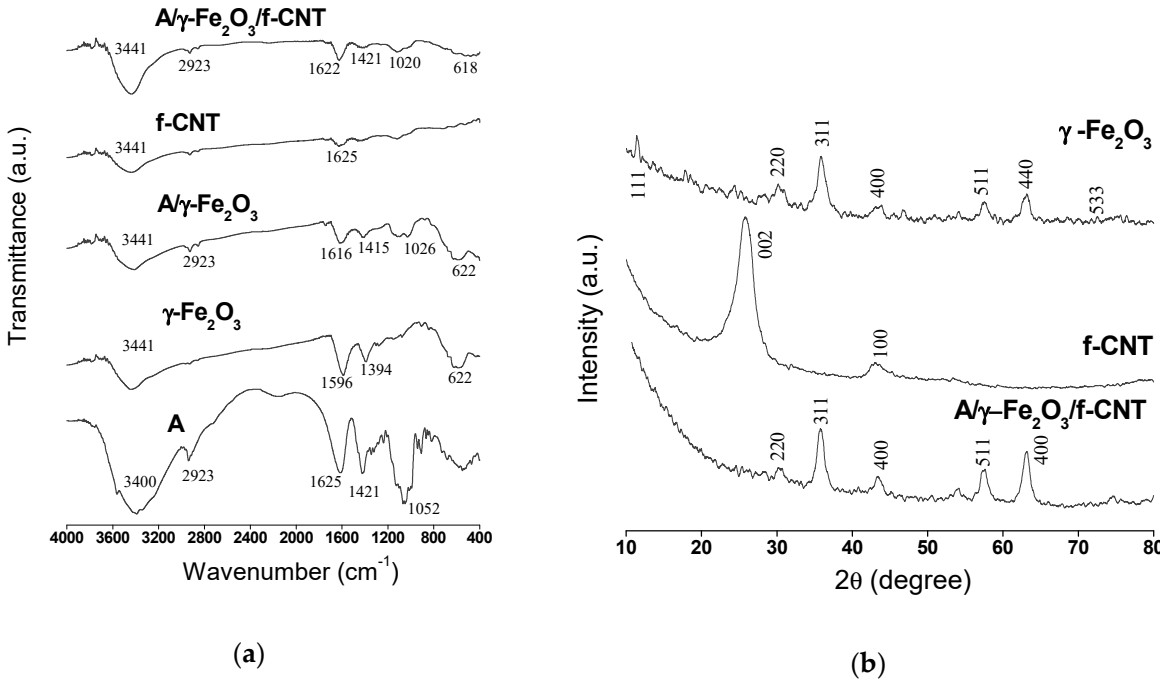

**(a)**　　　　　　　　　　　　　　　　　　　　　　**(b)**

**Figure 2.** (**a**) FTIR spectra of alginate (A), f-CNT, $\gamma$-Fe$_2$O$_3$ particles, A/$\gamma$-Fe$_2$O$_3$ and A/$\gamma$-Fe$_2$O$_3$/f-CNT beads; (**b**) XRD patterns of f-CNT, $\gamma$-Fe$_2$O$_3$ particles and A/$\gamma$-Fe$_2$O$_3$/f-CNT beads.

The XRD patterns of $\gamma$-Fe$_2$O$_3$, f-CNT and A/$\gamma$-Fe$_2$O$_3$/f-CNT materials can be seen in Figure 2b. The diffractograms obtained have broad diffraction lines, due to the nanometric size of $\gamma$-Fe$_2$O$_3$ and f-CNTs materials. For $\gamma$-Fe$_2$O$_3$, the peaks corresponding to the planes (111), (220), (311), (400), (511), (440) and (533) were observed at 2$\theta$ = 17.85°, 30.16°, 35.90°, 43.83°, 57.57°, 63.24° and 76.57°, respectively. The corresponding interplanar distance at 2$\theta$ = 43.83° is of d (400) = 0.2085 nm, that is very close to that reported in the literature [39,40]. The diffractogram fitted very well to the cubic symmetry of $\gamma$-Fe$_2$O$_3$ with the P4$_1$32 space group. The Rietveld analysis by the MAUD software confirmed the good crystallinity of $\gamma$-Fe$_2$O$_3$ with a cubic elementary mesh (a = 0.8341 nm). This value is in agreement with that reported in the literature for the same nanoparticles [41]. The average value of $\gamma$-Fe$_2$O$_3$ crystals

obtained by the refinement was of 10.59 nm, whereas the size calculated by Scherrer equation was of 10.24 nm. This value is in agreement with that obtained by Idris et al. [42], where resulted in 10 nm.

By comparison between the f-CNT diffractogram used in this study and that of the non-functionalized CNT reported in the literature [38], it could be concluded that the functionalization of CNTs with acid does not alter the crystallographic character of the material; only two peaks at $2\theta = 25.47°$ (002) and $2\theta = 42.81°$ (100) corresponding to the interplanar distances of 0.35 and 0.211 nm, respectively, were observed. The presence of $\gamma$-$Fe_2O_3$ in A/$\gamma$-$Fe_2O_3$/f-CNT was confirmed by the presence of the peaks attributed to the planes (220), (311), (400), (511) and (440) related to the interplanar distances of 0.294, 0.251, 0.209, 0.159 and 0.147 nm, respectively.

The characteristic peaks of f-CNT were not identified in the patterns of composite material (A/$\gamma$-$Fe_2O_3$/f-CNT) due to the low initial f-CNT weight ratio used in the preparation of alginate beads (0.4%). In addition, the presence of the peak at $2\theta = 43.83°$ (400) in maghemite XRD pattern caused the disappearance of the characteristic peak of f-CNT at $2\theta = 42.81°$ (100). These findings may be also attributed by the presence of alginate in the material, which reduces the purity of f-CNT in the final composite. It should be noted that A/$\gamma$-$Fe_2O_3$/f-CNT have the same characteristic peaks that $\gamma$-$Fe_2O_3$ have, indicating that the presence of alginate and f-CNTs in the composite did not alter the crystallographic character of $\gamma$-$Fe_2O_3$.

The thermal stability of the materials was evaluated by thermogravimetric analysis. Figure 3a-e show the TGA/TDG curves of the different samples accomplished in a temperature range of 30 to 900° C. The alginate curve showed an initial weight loss of 12.77 % between 30 and 200 °C corresponding to the loss of water. At temperature between 200 and 500 °C, the glycosidic bonds were destroyed which corresponds to a weight loss of 47.76 % [43]. The third loss of mass (16.71%) could be attributed to the formation of $CaCO_3$ [44]. Finally, the loss of mass of 8% corresponded to the decomposition of $CaCO_3$ which begins at 600 °C. Beyond 700 °C, only quicklime remained as the residue of the decomposition of $CaCO_3$, representing 14.75% of the initial mass of alginate. Thus, the total weight loss for alginate is 85.24%. One endothermic peak at 70°C and two exothermic peaks at 483°C and 533 °C were observed. The three peaks can be attributed to the dehydration of alginate, degradation of alginate and formation of $CaCO_3$, respectively. For $\gamma$-$Fe_2O_3$, the TGA/TDA (Figure 3b) curve suggested a weight loss of 8.16% at temperatures ranging from 30 to 240 °C, and 12.53% at temperatures from 240 to 475 °C. These mass losses could be ascribed to two endothermic peaks (at 89.27 and 305.4 °C) related to the evaporation of the water adsorbed on the surface of $\gamma$-$Fe_2O_3$ and to the degradation of organic matter resulting from the presence of citrate ions, respectively. The third loss in weight (16.08%) at 593.67 °C was attributed to the transition of $\gamma$-$Fe_2O_3$ to hematite, $\alpha$-$Fe_2O_3$ [45]. The total loss in mass for $\gamma$-$Fe_2O_3$ was of 36.77%. Above 845 °C, a $\gamma$-$Fe_2O_3$ mass weight of 63.22% remained. Likewise, the total weight loss of the f-CNT (Figure 3c) was only of 8.53%, with an endothermic peak at 85 °C. In the case of A/$\gamma$-$Fe_2O_3$ (Figure 3d), three losses in mass (9, 27.77 and 14.76%) were observed at 30-758 °C. These losses of mass are due to the loss of water, the thermal degradation of the alginate and citrates, the formation of $CaCO_3$ and the transition of $\gamma$-$Fe_2O_3$ to $\alpha$-$Fe_2O_3$, respectively. The presence of alginate in A/$\gamma$-$Fe_2O_3$ decreased the thermal stability of $\gamma$-$Fe_2O_3$, leading to a total weight loss of 51.53%. The residue corresponded to metal oxides and quicklime. The results showed that A/$\gamma$-$Fe_2O_3$ is stable at temperatures above 758 °C and that the material contains 48 ± 1 wt% of inorganic matter and 51 ± 1 wt% of organic matter, indicating that the mass ratio between inorganic and organic matter in A/$\gamma$-$Fe_2O_3$ is approximately of 1.0. In the case of A/$\gamma$-$Fe_2O_3$/f-CNT (Figure 3e), the weight loss is higher than that observed in the other materials. Four losses in mass were observed; the first (10.13%) corresponded to the dehydration of the beads; the second (11.29%) and the third (66.52%) attributed to the degradation of the organic matter and the fourth (3.35%) was due to the degradation of $CaCO_3$ and the transition of $\gamma$-$Fe_2O_3$ towards $\alpha$-$Fe_2O_3$. The organic matter content was of 91.29%, while the inorganic matter was of 8.43%. These mass losses corresponded to one exothermic peak at 202 °C and two endothermic peaks at 89 °C and 725 °C.

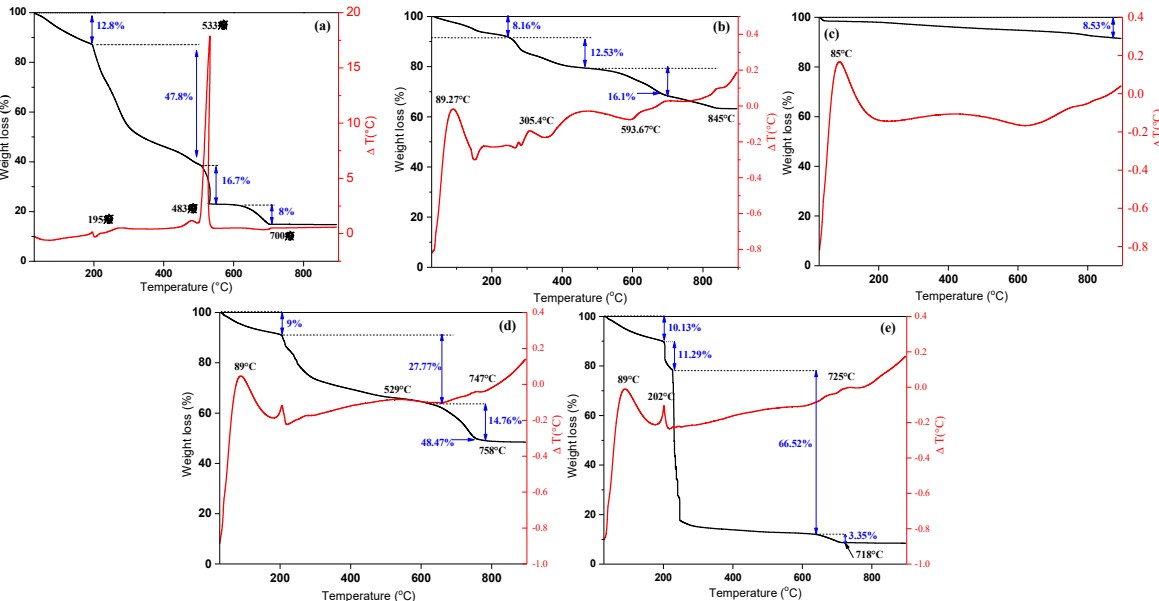

**Figure 3.** TGA/DTG profiles of (**a**) alginate, (**b**) γ-Fe$_2$O$_3$ particles, (**c**) f-CNT, (**d**) A/γ-Fe$_2$O$_3$, (**e**) A/γ-Fe$_2$O$_3$/f-CNT beads.

TGA/DTG profiles of γ-Fe$_2$O$_3$ particles, f-CNT, A/γ-Fe$_2$O$_3$ beads and A/γ-Fe$_2$O$_3$/f-CNT under air atmosphere were obtained (Figure S1a–d in Supplementary Material). For γ-Fe$_2$O$_3$, similar TG profile was observed (Figure S1a), with weight losses at the same temperatures that those detected under inert atmosphere. A great change was obtained for the TGA/DTG profile of f-CNT (Figure S1b), observing that practically the total mass of sample was burned off at 900 °C under air. The thermogravimetric profile of A/γ-Fe$_2$O$_3$ beads in oxidizing atmosphere (Figure S1c) was similar to that obtained under inert conditions, with a more intense decay in the weight sample at approximately 200 °C. Finally, TGA/DTG curves of A/γ-Fe$_2$O$_3$/f-CNT composite (Figure S1d) showed a dramatically decreasing in the weight of sample at ~250 °C, remaining a weight content of 7.37% at the final temperature, attributed to the iron oxides amount present in the composite.

SEM micrographs of the synthesized materials are shown in Figure 4a–e. A TEM micrograph of f-CNT material is shown in Figure S2 (Supplementary Material). The SEM images showed the roughness of the surface of alginate beads (Figure 4a). For carbon nanotubes, the cylindrical form of f-CNTs could be appreciated with a diameter from 20 to 50 nm and a length of 50-100 nm (Figure 4b, Figure S2). γ-Fe$_2$O$_3$ nanoparticles are of spherical shape, showing many aggregations due to the drying before SEM analysis (Figure 4c). The elemental composition of γ-Fe$_2$O$_3$ obtained by EDX analysis was as follows: %C of 19.96, %O of 25.15, %Na of 7.92 and %Fe of 46.97%, confirming the coating of the particles with sodium citrates. The nanoparticles showed a larger size than that determined by XRD; this is due to the aggregation of nanoparticles because of van der Waals forces and magnetic repulsions [46]. For A/γ-Fe$_2$O$_3$ beads (Figure 4d), their external surface appeared very rough with a uniform dispersion of γ-Fe$_2$O$_3$ nanoparticles. Finally, in the case of A/γ-Fe$_2$O$_3$/f-CNT (Figure 4e), the surface is very rough and heterogeneous, observing areas with larger aggregations than those observed in A/γ-Fe$_2$O$_3$ beads. The coupling of f-CNTs with γ-Fe$_2$O$_3$ nanoparticles in the alginate matrix lead to improve the mechanical properties of the composite and to have aggregations in the beads [47].

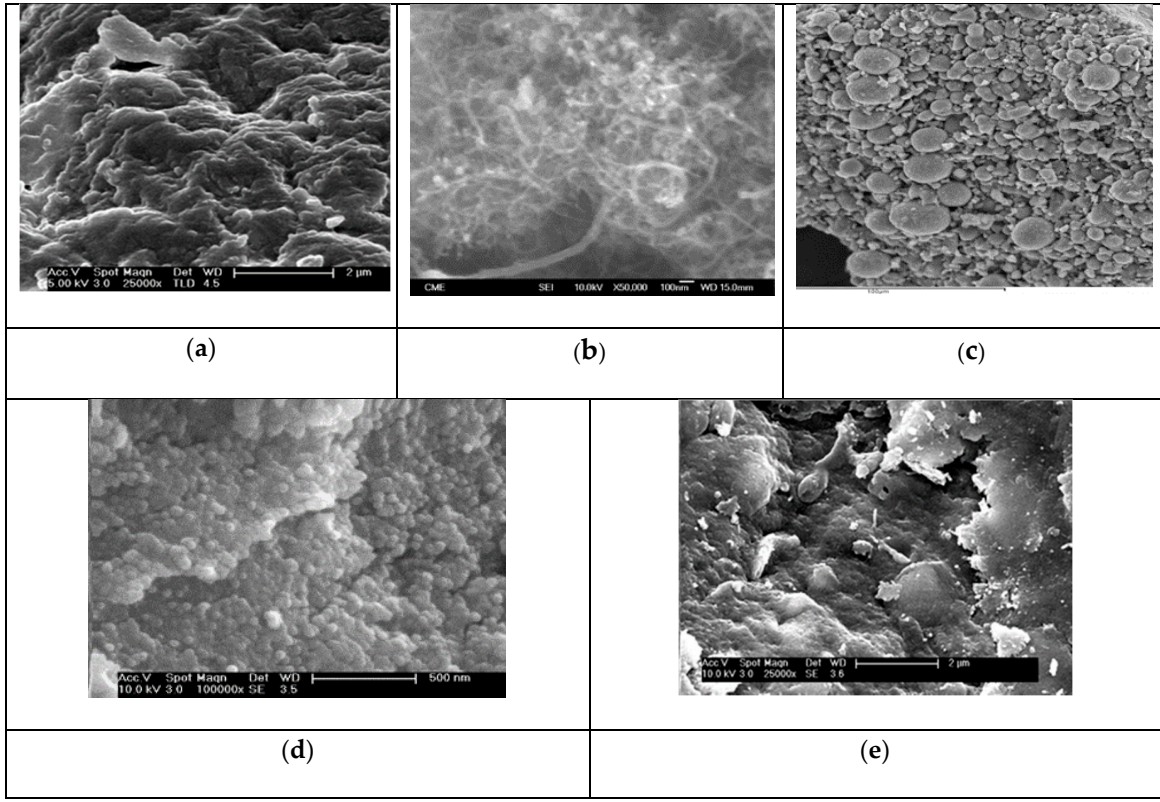

**Figure 4.** SEM micrographs of (**a**) alginate, (**b**) f-CNT, (**c**) $\gamma$-Fe$_2$O$_3$ particles, (**d**) A/$\gamma$-Fe$_2$O$_3$ and (**e**) A/$\gamma$-Fe$_2$O$_3$/f-CNT beads.

The magnetization curves of $\gamma$-Fe$_2$O$_3$, A/$\gamma$-Fe$_2$O$_3$ and A/$\gamma$-Fe$_2$O$_3$/f-CNTs materials are shown in Figure 5. The superposition of the curves indicated that $\gamma$-Fe$_2$O$_3$ nanoparticles maintained their magnetic properties when they were encapsulated in the beads. According to the shape of the obtained curves, the materials have a magnetic memory with superparamagnetic behavior. The magnetization saturation values, determined at 295 K, resulted in 42.63, 29.06 and 27.16 emu g $^{-1}$ for $\gamma$- Fe$_2$O$_3$, A/$\gamma$-Fe$_2$O$_3$ and A/$\gamma$-Fe$_2$O$_3$/f-CNT, respectively.

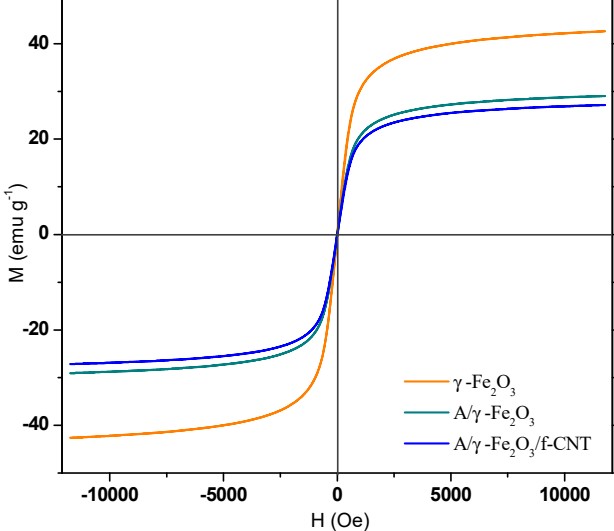

**Figure 5.** Magnetization hysteresis loop of $\gamma$-Fe$_2$O$_3$ particles, A/$\gamma$-Fe$_2$O$_3$ and A/$\gamma$-Fe$_2$O$_3$/f-CNT beads.

The magnetization saturation of A/γ-Fe$_2$O$_3$/f-CNT beads was slightly lower than that of A/γ-Fe$_2$O$_3$ beads, but it stills enough to be efficiently separated from the aqueous medium. From these magnetization measurements, the mass fractions of γ-Fe$_2$O$_3$ nanoparticles in A/γ-Fe$_2$O$_3$ and A/γ-Fe$_2$O$_3$/f-CNT beads were estimated to be 65 and 60%, respectively. These results revealed that the synthesized magnetic beads are excellent materials for wastewater treatment in large-scale applications because they can be easily separated from water using a magnetic field.

### 3.2. Effect of Initial PH in Adsorption Tests

The pH of the solution is an important factor that could affect the adsorption process by controlling both the surface chemistry of the adsorbent and the speciation of the adsorbate. The experiments were carried out in a pH range of 3.0–10.0 and the results are reported in Figure 6. It could be observed that MB uptake increased from 37.1 to 82.7 mg g$^{-1}$ with increasing the pH solution from 3 to 10. These results are directly related to the chemical surface properties of the adsorbent and MB molecules.

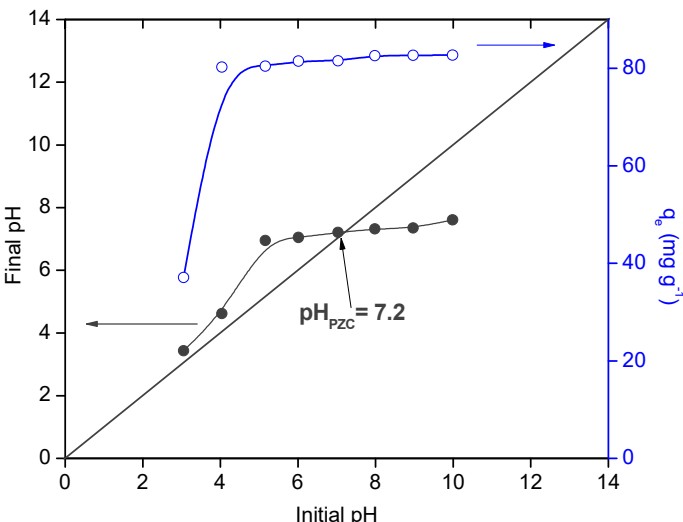

**Figure 6.** Effect of Ph solution on methylene blue (MB) equilibrium adsorption capacity.

Thus, the point of zero charge (Ph$_{PZC}$) of the beads was found to be 7.2 in MB solution [48]. The results showed that a remarkable decreasing of the adsorption capacity was observed at Ph values < Ph$_{PZC}$, related to the pKa values of alginate (3.38 and 3.65) [46]. At Ph values higher than 3.65, more negative charges are generated on the surface of the beads, while MB molecules showed positive charge, thus promoting the adsorbate-adsorbent electrostatic interactions between the negatively charged groups (–COOH and –OH) in the beads surface and the positively charged molecules of MB. In addition, at Ph < pKa values of alginate, other interactions such as hydrogen bonding between sulfur and/or nitrogen atoms in MB and protonated carboxylate groups might be dominant in the adsorption process [34]. Same results were found in the literature for MB adsorption onto other composite beads [5–16,49].

### 3.3. Kinetic Studies

Figure 7 shows the kinetic adsorption curve of MB adsorption onto A/γ-Fe$_2$O$_3$/f-CNT composite beads. Adsorption increased with increasing contact time due to the abundant availability of active sites on the adsorbent structure [50]. After, these sites were sequentially occupied and the adsorption capacity assume less valuable. In this case, the equilibrium time was achieved within 48 h. The experimental kinetic data were fitted to the non-linear form of pseudo-first order (PFO) and pseudo-second order (PSO) models (Equations (5) and (6), respectively) [51,52]:

$$q_t = q_e(1 - e^{-k_1 t}) \tag{5}$$

$$q_t = \frac{k_2 q_e^2 t}{1 + k_2 q_e t} \tag{6}$$

where, $q_e$ and $q_t$ (mg g$^{-1}$) are the adsorption capacities at equilibrium and at any time, respectively; and $k_1$ (min$^{-1}$) and $k_2$ (g mg$^{-1}$ min$^{-1}$) are the adsorption rate constants of the PFO and PSO models, respectively. The fitting of the experimental data to both models is illustrated in Figure 7. According to the calculated values of $R^2$ and RMSE, it could be concluded that the adsorption process was well-fitted by the PSO model, obtaining a rate constant value of 0.030 g mg$^{-1}$ min$^{-1}$.

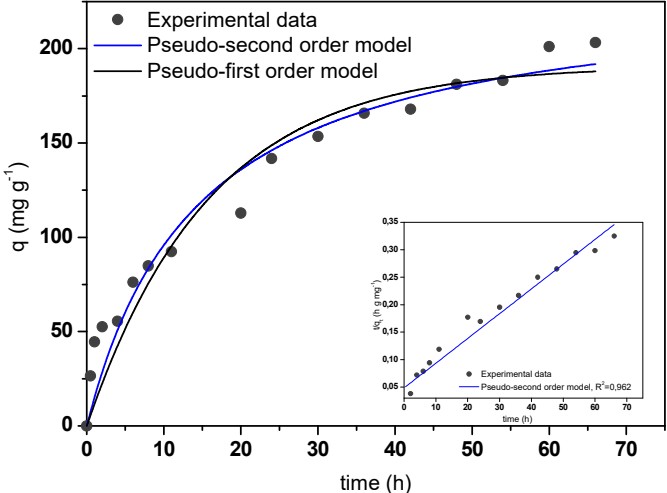

**Figure 7.** Experimental and theoretical kinetic curve of MB adsorption onto A/γ-Fe$_2$O$_3$/f-CNT beads.

### 3.4. Adsorption Isotherm Studies

The MB adsorption isotherm data were evaluated by three adsorption isotherm equations, e.g., Langmuir, Freundlich and Sips models [53–55]. Langmuir model is related to a homogeneous monolayer adsorption process, while Freundlich model is widely used for heterogeneous multilayer adsorption. Overall, Sips model is a combination of both Langmuir and Freundlich equations. Langmuir, Freundlich and Sips models can be expressed as follows (Equations (7)–(9), respectively):

$$q_e = \frac{q_m K_L C_e}{1 + K_L C_e} \tag{7}$$

$$q_e = K_F C_e^{1/n} \tag{8}$$

$$q_e = \frac{q_m \left(b_s C_e^{1/s}\right)}{1 + \left(b_s C_e^{1/s}\right)} \tag{9}$$

where, $q_e$ (mg g$^{-1}$) is the equilibrium adsorption capacity and $C_e$ (mg L$^{-1}$) is the equilibrium MB concentration in solution; $q_m$ (mg g$^{-1}$) is the maximum monolayer capacity and $K_L$ (L mg$^{-1}$) is the Langmuir constant, indicative of the affinity of the adsorbent towards the pollutant. Likewise, $K_F$ (L g$^{-1}$) and $1/n$ are parameters of Freundlich equation; the value of $1/n$ is representative of the adsorption intensity. Finally, $b_s$ (L mg$^{-1}$) is a constant of Sips equation, which depends on the temperature, and $1/s$ is another parameter related to the adsorption energy.

Experimental and theoretical MB adsorption data are shown in Figure 8 and the calculated adsorption equilibrium parameters are collected in Table 2. According to the obtained low RMSE and high $R^2$ values, it could be affirmed that Sips model described very satisfactorily the experimental MB equilibrium adsorption data. Thus, the maximum adsorption capacity calculated from the model (320.5 mg g$^{-1}$) was found very close to the experimental value (307.0 mg g$^{-1}$), suggesting the homogenous surface of the adsorbent and the formation of a monolayer of MB molecules on its surface

in the studied range of MB concentration (5–270 mg L$^{-1}$). In this study, 1/n value resulted in 0.47, lower than 1.0, indicative of a favorable adsorption process.

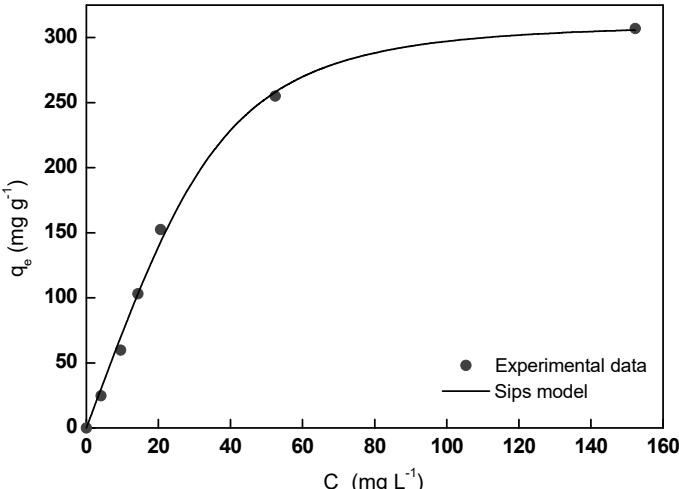

**Figure 8.** Experimental and theoretical MB adsorption isotherms onto A/$\gamma$-Fe$_2$O$_3$/f-CNT beads.

**Table 2.** MB adsorption isotherm parameters onto A/$\gamma$-Fe$_2$O$_3$/f-CNT beads at room temperature.

| Model | Parameter | Value |
|---|---|---|
| **Langmuir** | $K_L$ (L mg$^{-1}$) | 0.03 |
| | $q_{max}$ (mg g$^{-1}$) | 396.7 |
| | $R^2$ | 0.977 |
| | RMSE | 17.4 |
| **Freundlich** | $K_F$ (L g$^{-1}$) | 31.6 |
| | 1/n | 0.5 |
| | $R^2$ | 0.907 |
| | RMSE | 35.5 |
| **Sips** | $q_{max}$ (mg g$^{-1}$) | 320.5 |
| | $K_s$ (L mg$^{-1}$) $\times 10^3$ | 6.4 |
| | $n_s$ | 0.6 |
| | $R^2$ | 0.999 |
| | RMSE | 4.1 |

The maximum MB adsorption capacity values on several adsorbents reported in the literature are collected in Table 3. The significant adsorption capacity of A/$\gamma$-Fe$_2$O$_3$/f-CNT revealed that this composite is an excellent adsorbent for the removal of MB contaminant from wastewater.

**Table 3.** Comparison of several MB adsorption capacities reported in the literature.

| Adsorbent | $C_0$ (mg $L^{-1}$) | $q_{max}$ (mg $g^{-1}$) | Model | Reference |
|---|---|---|---|---|
| Mesoporous activated carbon-alginate beads | 20–100 | 230.0 | Freundlich | [29] |
| Iron oxide/carbon nanocomposites | | 118.2 | | [16] |
| Poly(vinyl alcohol)-sodium alginate-chitosan-montmorillonite hydrogel beads | | 137.2 | Freundlich | [56] |
| Chitosan-g-poly(acrylic acid) hydrogel modified with cellulose nanowhiskers | | 1968 | | [57] |
| Alginate-halloysite nanotube beads | | 250.0 | | [58] |
| MWCNTs untreated | | 132.6 | | [59] |
| MWCNTs alkali-activated | | 399.0 | | [60] |
| Alginate-bentonite-activated carbon (ABA) | 25–500 | 756.9 | Freundlich | [5] |
| Alginate/acid-activated organobentonite beads (A-OAB) | 20–500 | 799.4 | Langmuir | [33] |
| Alginate (A) | 20–500 | 483.6 | Chapman | [33] |
| Functionalized MWCNTs | 20–1000 | 260.8 | Freundlich | [61] |
| $\gamma$-$Fe_2O_3$ | 20–1000 | 155.3 | Langmuir | [61] |
| Alginate/$\gamma$-$Fe_2O_3$ | 20–1000 | 438.6 | Langmuir | [61] |
| Alginate/maghemite/functionalized multiwalled carbon nanotubes beads (A/$\gamma$-$Fe_2O_3$/f-CNTs) | 5–270 | 320.5 | Langmuir | This work |

*3.5. Statistical Physics Analysis (Monolayer Model)*

The monolayer model [60] can be considered as the general case of Langmuir model. It is applied to describe an entire set of experimental data. The monolayer model with single energy assumes that a site can accept "n" number of molecules. The equation of this model is:

$$q_e = \frac{n \cdot N_M}{1 + \left(\frac{C_{1/2}}{C_e}\right)^n}$$

(10)

where, n is the number of molecules fixed per adsorbent surface, $C_{1/2}$ (mg $L^{-1}$) is the concentration at half saturation, $N_M$ (mg $g^{-1}$) is the receptor site density effectively occupied, $q_e$ (mg $g^{-1}$) and $C_e$ (mg $L^{-1}$) are the MB adsorption capacity and concentration, respectively, at equilibrium time.

According to the study reported by Meili et al. [62], if the number of dye molecules is lower than 1.0, it indicates the multi-anchorage adsorption process, which reflects a parallel position of the dye molecule adsorbed on the surface. This means that the dye molecules can simultaneously interact with several functional groups present on the adsorbent surface. On the contrary, if n value is higher than 1.0, adsorption can be considered as a multi-molecular process in which the molecules can interact in vertical position with the functional groups of the beads surface.

In this study, n parameter took a value of 1.62. Hence, the adsorption process can be considered as a multi-molecular process, assuming a vertical geometry of the dye molecule in the adsorption process. Based on this model, it is possible to characterize the interactions between the MB molecules and the beads surface by the adsorption energy determination. The energetic parameter can be calculated from:

$$\Delta E = R \cdot T \cdot \ln\left(\frac{C_s}{C_{1/2}}\right)$$

(11)

where, $C_s$ (mg $L^{-1}$) is the solubility of MB in water and $C_{1/2}$ (mg $L^{-1}$) is the half saturation concentration determined by the fitting method.

The value of the energy of adsorption was found of 19.1 kJ $mol^{-1}$, suggesting that the adsorption is of endothermic and physical nature. The same result was obtained by Meili et al. [62] for the adsorption of MB onto an agricultural Algerian olive cake waste.

*3.6. Thermodynamic Adsorption*

The effect of the temperature on MB adsorption onto A/$\gamma$-$Fe_2O_3$/f-CNT composite beads was studied at four different values, i.e., 283, 293, 303 and 313 K. The results showed a positive correlation between MB adsorption and temperature, suggesting the endothermic nature of the process, which is in agreement with the results obtained from the monolayer model. The thermodynamic parameters, e.g., the enthalpy or heat of adsorption ($\Delta H^\circ$), the change in the standard Gibbs' energy ($\Delta G^\circ$) and entropy ($\Delta S^\circ$) were calculated using the following equations [27]:

$$\Delta G^\circ = -R \cdot T \cdot \ln K_c$$

(12)

$$K_c = \frac{q_e}{C_e}$$

(13)

$$\log\left(\frac{q_e}{C_e}\right) = \frac{-\Delta H^\circ}{2.303 \cdot R \cdot T} + \frac{\Delta S^\circ}{2.303 \cdot R}$$

(14)

where, $q_e$ (mg $g^{-1}$) is the equilibrium MB adsorption capacity, $C_e$ (mg $L^{-1}$) is the equilibrium MB concentration, $\Delta H^\circ$ (kJ $mol^{-1}$) is the enthalpy variation, $\Delta S^\circ$ (J $mol^{-1}$ $K^{-1}$) is the entropy, $\Delta G^\circ$ (kJ $mol^{-1}$) is the change in the standard Gibbs' energy, $K_c$ is the equilibrium constant, T (K) is the temperature and R is the universal gas constant (8.314 J $mol^{-1}$ $K^{-1}$). The enthalpy ($\Delta H^\circ$) and entropy ($\Delta S^\circ$)

parameters were evaluated by plotting $\log(q_e/C_e)$ versus $1/T$, resulting in values of 36.9 kJ mol$^{-1}$ and 195.1 J mol$^{-1}$ K$^{-1}$, respectively.

The obtained positive value of the enthalpy is indicative of an endothermic process. Thus, the enthalpy value resulted smaller than 40 kJ mol$^{-1}$, indicating that MB adsorption onto A/$\gamma$-Fe$_2$O$_3$/f-CNT composite beads can be considered as a physisorption process, governed by electrostatic interactions, mainly van der Waals forces. $\Delta G°$ parameters at 283, 293, 303 and 313 K were of $-18.3$, $-20.2$, $-22.2$ and $-24.1$ kJ mol$^{-1}$, respectively. The negative $\Delta G°$ values indicated the spontaneous and the physical nature of MB adsorption process onto the synthesized composites. The change in the adsorption entropy ($\Delta S°$) was positive, suggesting that the MB molecules increased their randomness at the solid/liquid interface during the adsorption.

### 3.7. Proposed Adsorption Mechanism of MB onto A/$\gamma$-Fe$_2$O$_3$/f-CNT Composite Beads

The adsorption mechanism was investigated by FTIR analysis. The FTIR spectra of MB and A/$\gamma$-Fe$_2$O$_3$/f-CNT composite before and after adsorption are depicted in Figure 9. The adsorption mechanism of methylene blue has been extensively reported in the literature. Thus, Wang et al. [63] have previously found that MB molecules could be adsorbed on alginate and acidified MWCNTs beads, on one hand, by $\pi$-$\pi$ dispersion interactions with the aromatic regions of MWCNTs, and, on the other hand, by electrostatic attractions with the carboxylate groups present in alginate and MWCNTs. So, different adsorption mechanisms can be proposed when the adsorbent is a composite formed by different materials. The interactions between MB molecule and A/$\gamma$-Fe$_2$O$_3$/f-CNTs beads were verified by FTIR analysis (Figure 9).

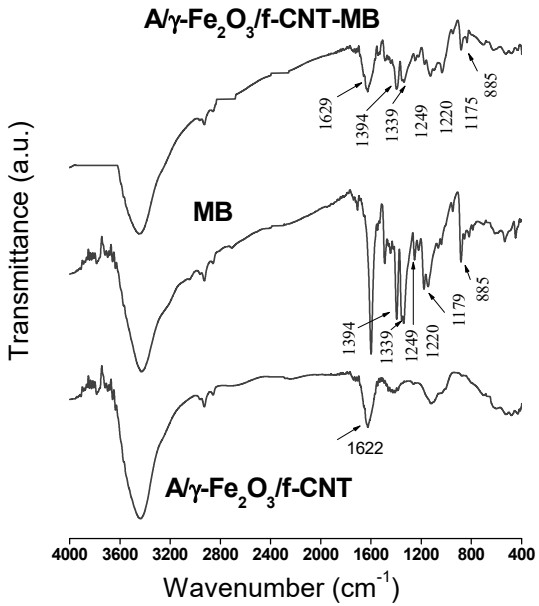

**Figure 9.** FTIR spectra of MB and A/$\gamma$-Fe$_2$O$_3$/f-CNT beads before and after MB adsorption.

After adsorption, the FTIR spectrum of A/$\gamma$-Fe$_2$O$_3$/f-CNT clearly showed the MB signature, indicating its adsorption onto the composite beads. The bands appearing at 1394 and 1339 cm$^{-1}$ are referred to the aromatic rings of MB. The characteristic C=C band of the aromatic rings detected at 1179 cm$^{-1}$ in the MB spectrum appeared at 1175 cm$^{-1}$ in the FTIR of A/$\gamma$-Fe$_2$O$_3$/f-CNT after adsorption [64]. In addition, the C-H bond vibrations of MB at 885, 1220 and 1249 cm$^{-1}$ in A/$\gamma$-Fe$_2$O$_3$/f-CNT/MB were observed [65]. A shift in frequency values from 1622 to 1629 cm$^{-1}$ for the carboxylate and carbonyl groups of A/$\gamma$-Fe$_2$O$_3$/f-CNT beads after adsorption was determined. Variation in intensity, shift and occurrence of new vibrations in A/$\gamma$-Fe$_2$O$_3$/f-CNTs-MB suggested that the mechanism of MB uptake by A/$\gamma$-Fe$_2$O$_3$/f-CNTs beads may be controlled by electrostatic interactions involving the functional groups

present in the beads. These results are in agreement with the studies of pH and temperature effects on MB adsorption.

### 3.8. Adsorbent Recycling

As it was elucidated in the pH effect adsorption studies, A/$\gamma$-Fe$_2$O$_3$/f-CNT composite beads seem to be ineffective for MB removal at pH values lower than 3, indicating a high possibility that MB adsorbed ions be desorbed in acidic medium (0.1M HNO$_3$ solution). Desorption efficiencies of the accomplished consecutive adsorption-desorption cycles are depicted in Figure 10.

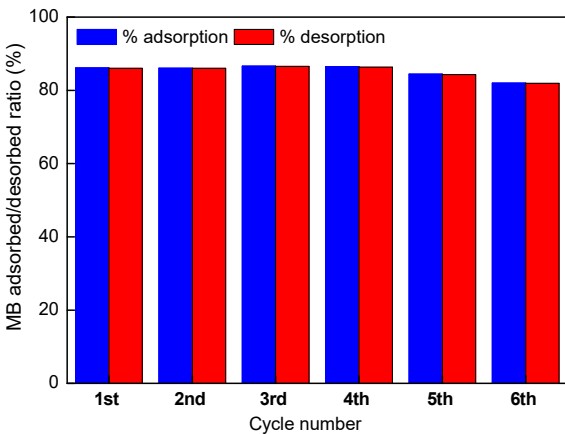

**Figure 10.** Desorption efficiency of MB onto A/$\gamma$-Fe$_2$O$_3$/f-CNT composites after 6 adsorption-desorption cycles.

At low pH values, the functional groups present on the beads surface are protonated, which leads to reduce the electrostatic attractions between MB molecules and the adsorbent surface. This conduct to the diffusion of MB molecules to the acid solution. As it could be observed in the Figure, 6 consecutive adsorption-desorption tests using a nitric acid solution as elution reagent were assessed. Thus, A/$\gamma$-Fe$_2$O$_3$/f-CNT composite beads could be reused for 6 cycles maintaining its high adsorptive ability, observing similar MB adsorption/desorption ratio, ranging from 82% to 87%. The results confirmed that A/$\gamma$-Fe$_2$O$_3$/f-CNT beads are efficient adsorbents of methylene blue, since they can be rapidly and effectively regenerated. Same findings have been described in the literature [66].

## 4. Conclusions

Magnetic beads of alginate citrate-stabilized maghemite nanoparticles and functionalized multiwalled carbon nanotubes (A/$\gamma$-Fe$_2$O$_3$/f-CNT) were prepared, characterized and used as adsorbent for methylene blue dye. The kinetic and equilibrium experimental data were well-explained by the pseudo-second order kinetic model and Sips isotherm equation, respectively. The maximum adsorption capacity determined by Sips model was of 320.5 mg g$^{-1}$, very close to the experimental value (307.0 mg g$^{-1}$), suggesting a homogenous adsorption process in the studied range of MB concentrations. The studies of pH and temperature effect on the adsorption process revealed that the beads exhibited an important adsorptive stability over a large range of pH values (4–10) and the endothermic and spontaneous nature of the process. The statistical study using the monolayer model demonstrated the multi-molecular nature of the adsorption process, with a vertical adsorption geometry of the dye. The magnetic beads could be used at least for six adsorption/desorption successive cycles, maintaining its adsorptive capacity. According to the results of this study, the synthesized composite beads can be considered as high effective adsorbents of the model compound, revealing as excellent materials for dyes removal in flow-through adsorption WWTP applications.

**Supplementary Materials:** The supplementary materials are available online at http://www.mdpi.com/2076-3417/9/21/4563/s1.

**Author Contributions:** Conceptualization, S.A.-T., M.B., and N.B.; data curation, M.B., and N.B.; formal analysis, S.A.-T., M.B., and N.B.; investigation, S.A.-T., M.B., N.B., and M.M.; methodology, S.A.-T., M.B., and N.B.; project administration, M.B., and N.B.; resources, S.A.-T., M.B., N.B., and M.M.; software, S.A.-T., M.B., N.B., and M.M.; supervision, S.A.-T., and M.B.; writing-original draft, M.B.; Writing-review & Editing, S.A.-T., M.B., and N.B.

**Funding:** This research has not received any specific grant from funding agencies in the public, commercial, or not-for-profit sectors.

**Acknowledgments:** The authors wish to thank University Ferhat Abbas of Setif, Algeria, for the financial support. Additionally, the authors acknowledge the scientific and technical support from Autónoma University and Complutense University of Madrid, Spain.

**Conflicts of Interest:** The author(s) declared no potential conflicts of interest with respect to the research, authorship, and/or publication of this article.

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
