# Peer review of "Effective Adsorption of Methylene Blue dye onto Magnetic Nanocomposites. Modeling and Reuse Studies"

_applsci, doi:10.3390/app9214563_

Round 1

Reviewer 1 Report

The authors reported the adsorption of MB from aqueous solution using alginate and maghemite nanoparticles functionalized CNTs in their manuscript titled “Effective Adsorption of Methylene Blue dye onto magnetic nanocomposites. Modeling and reuse studies”. The research is normal routine work and similar investigations for the adsorption of MB have been reported previously on numerous occasions. The preparation of the adsorbents is the part that could impart new knowledge in the field. I recommend the acceptance after addressing the minor revision of the manuscript by addressing the following points;

In the introduction, it is claimed that activated carbons are high-cost material but how come nanomaterials such as the ones reported in this study be cost-effective when compared to activated carbons? Making a combination of trio (alginate/Fe2O3 and CNT) could prove more expensive than the activated carbon prepared from a single precursor. Avoid using the word “high” for specific surface area. These are not high specific surface area materials. What is the reason behind the disappearance of the two XRD peaks of f-CNT in the composite A/γ-Fe2O3/f-CNT? The TG/TDG section is confusing. As per the figure caption, b) is f-CNT and C0 is γ-Fe2O3 but their explanation is in reverse order on page 7. For example, “For γFe2O3, the TGA/TDA (Fig.3b) curve suggested a weight loss of 8.16% at temperatures ranging from 30 to 240 °C, and 12.53%.” and the figure caption says 3b is f-CNT. Same with Fig 3c. It needs to be corrected. It seems that some parts in equations 7 and 9 are written in some language other than English. Please correct.

Author Response

Dear Ms. Allen Dou

Assistant Editor,

Applied Sciences

Thanks for your letter and for sending us the referee comments on the manuscript “Effective adsorption of Methylene Blue dye onto magnetic nanocomposites. Modeling and reuse studies” by S. Álvarez-Torrellas, M. Boutahala, N. Boukhalfa and M. Munoz (applsci-609030) that we sent to be considered for publication in Applied Sciences. Please, enclose you can find the revised version of our manuscript.

We are pleased to find that the referees highlighted the suitability of paper for publication after some revision:

Referee 1. “The preparation of the adsorbents is the part that could impart new knowledge in the field. I recommend the acceptance after addressing minor revision”…

Referee 2. “However, the second part of the paper (modeling) is an important work so the paper can be published”…

Referee 3. “The results were well presented”….

General remarks:

We have changed the manuscript according to the suggestions and comments of the referees, which makes the manuscript more interesting and informative. We acknowledge the remarks from the reviewers which helped us to improve the quality and the comprehension of the work.

We have included some new references, in accordance with the suggestions of the referees. Additionally, SEM technique have been used in order to improve the exploring of the morphological properties of carbon nanotubes; and TGA/DTG analysis of the materials under air atmosphere have been carried out.

We submit a copy of the final version of the manuscript, with the changes marked up in red.

The whole manuscript was edited for proper English language, grammar, punctuation, spelling, and overall style by one qualified native English. On the other hand, the references are formatted according to the requirements of Applied Sciences.

We think that these comments support our (already comprehensive) descriptions, as we explain in the appended report.

Yours sincerely,

Prof. Silvia Álvarez-Torrellas and Prof. Moktar Boutahala.

Detailed comments and list of changes:

Effective adsorption of Methylene Blue dye onto magnetic nanocomposites. Modeling and reuse studies” by S. Álvarez-Torrellas et al. (applsci-609030)

Referee 1

Thanks for your comments on the manuscript. We acknowledge the remarks which helped us to improve the quality and the comprehension of the work.

Reviewer comments

The authors reported the adsorption of MB from aqueous solution using alginate and maghemite nanoparticles functionalized CNTs in their manuscript entitled “Effective adsorption of Methylene Blue dye onto magnetic nanocomposites. Modeling and reuse studies”. The research is normal routine work and similar investigation for the adsorption of MB have been reported previously on numerous occasions. The preparation of the adsorbents is the part that could impart new knowledge in the field. I recommend the acceptance after addressing the minor revision of the manuscript by addressing the following points;

In the Introduction, it is claimed that activated carbons are high-cost material but how come nanomaterials such as the ones reported in this study be cost-effective when compared to activated carbons? Making a combination of trio (alginate/Fe2O3 and CNT) could prove to more expensive than the activated carbon prepared from a single precursor.

REPLY and MODIFICATION

We thank the referee for this suggestion. We are agree with the referee. The manuscript, Chapter 1. Introduction, has been amended as follows: “Conventional activated carbon materials are commonly used as adsorbents; however, the difficulties for the separation after treatment in some applications limit their use and, in some cases, make the process unfeasible for industrial applications [7-8].

As it has been widely reported in the literature, conventional activated carbons, synthesized from a single precursor, are economical materials used in the adsorbent processes, and even finding specific applications, such as supercapacitors [9] and catalysts [10], due to their relatively low cost”.

“…the use of MWCNTs as nano-adsorbents is limited due to their separation and regeneration difficulties”.

In the Results and Discussion Chapter, 3.8. Adsorbent recycling, the manuscript has been amended as follows: “The results confirmed that A/γ-Fe2O3/f-CNT beads are efficient adsorbents of aromatic compounds from wastewater, since they can be rapidly and effectively regenerated”.

Gomri, F.; Finqueneisel, G.; Zimny, T.; Korili, S.A.; Gil, A.; Boutahala, M. Adsorption of Rhodamine 6G and humic acids on composite bentonite–alginate in single and binary systems. Appl. Water Sci. 2018, 8, https://doi.org/10.1007/s1320. Oladipo, A.A.; Ifebajo, A.O. Highly efficient magnetic chicken bone biochar for removal of tetracycline and fluorescent dye from wastewater: Two-stage adsorber analysis. J. Environ. Manage. 2018, 209, 9-16. Ruiz, V.; Blanco, C.; Santamaría, R.; Ramos-Fernández, J.M.; Martínez-Escandell, M.; Sepúlveda-Escribano, A.; Rodríguez-Reinoso, F. An activated carbon monolith as an electrode material for supercapacitors. Carbon 2009, 47, 195-200. Rodríguez-Reinoso, F. The role of carbon materials in heterogeneous catalysis. Carbon 1998, 36, 159-175.

Reviewer comments

Avoid using the word “high” for specific surface area. These are not high specific surface area materials.

REPLY and MODIFICATION

We thank the referee for this suggestion. We are agree with the referee.

The Abstract of the manuscript has been amended as follows: “The beads (A/γ-Fe2O3/f-CNT) presented a relatively low BET specific surface area value of 59 m2.g-1.”

The Introduction Section has been amended: “These oxide materials demonstrated a huge potential for their application in wastewater treatment due to their, magnetic properties, relevant selectivity, reactivity and biocompatibility abilities”.

In the Results and Discussion, attending to the results shown in Figure 1 and Table 1, the manuscript has been amended: “Thus, f-CNT material showed a low-moderate specific surface area (265 m2 g-1), since relatively low BET area values were found for γ-Fe2O3 nanoparticles and A/γ-Fe2O3/f-CNT composites, 35 and 59 m2 g-1, respectively”.

Reviewer comments

What is the reason behind the disappearance of the two XRD peaks of f-CNT in the composite A/γ-Fe2O3/f-CNT?

REPLY and MODIFICATION

We agree with the referee. The manuscript has been amended as follows: “The characteristic peaks of f-CNT were not identified in the patterns of composite material (A/γ-Fe2O3/f-CNT) due to the low initial f-CNT weight ratio used in the preparation of alginate beads (0.4%). In addition, the presence of the peak at 2θ = 43.83o (400) in maghemite XRD pattern caused the disappearance of the characteristic peak of f-CNT at 2θ = 42.81° (100). These findings may be also attributed by the presence of alginate in the material, which reduces the purity of f-CNT in the final composite”.

Reviewer comments

The TG/TDG section is confusing. As per the figure caption, b) is f-CNT and C0 is γ-Fe2O3 but their explanation is in reverse order on page 7. For example, “For γFe2O3, the TGA/TDA (Fig.3b) curve suggested a weight loss of 8.16% at temperatures ranging from 30 to 240 °C, and 12.53%.” and the figure caption says 3b is f-CNT. Same with Fig 3c. It needs to be corrected.

REPLY and MODIFICATION

We thank the referee for this comment. We are agree with the referee. The discussion is correct but there was a mistake in the figure caption of Figure 3. The manuscript has been amended: “Figure 3. TGA/DTG profiles of (a) alginate, (b) γ-Fe2O3 particles, (c) f-CNT, (d) A/γ-Fe2O3, (e) A/γ-Fe2O3/f-CNT beads”.

Reviewer comments

It seems that some parts in equations 7 and 9 are written in some language other than English. Please correct.

REPLY and MODIFICATION

We thank the referee for this suggestion. The manuscript has been amended as follows: “where, qe (mg g-1) is the equilibrium adsorption capacity and Ce (mg L-1) is the equilibrium MB concentration in solution; qm (mg g-1) is the maximum monolayer capacity and KL (L mg-1) is the Langmuir constant, indicative of the affinity of the adsorbent towards the pollutant. Likewise, KF (L g-1) and 1/n are parameters of Freundlich equation; the value of 1/n is representative of the adsorption intensity. Finally, bs (L mg-1) is a constant of Sips equation, which depends on the temperature, and 1/s is another parameter related to the adsorption energy”.

Dr. Silvia Alvarez-Torrellas

Assistant Professor in Chemical Engineering

Complutense University

Department of Chemical Engineering and Materials. Fac. C. C. Químicas.

Avda. Complutense s/n, 28040 Madrid-Spain, Tel: +34 91 3944118

Dr. Moktar Boutahala

Université Ferhat Abbas

Laboratoire de Génie des Procédés Chimiques (L.G.P.C.)

Département de Génie des Procédés, Faculté de Technologie

Sétif 19000, Algeria

Reviewer 2 Report

The paper "Effective adsorption...." by S. Alvarez-Torrellas is focused in the adsorption of methylene blue onto magnetic nanocomposites. The adsorption of metals, VOCs... in aqueous media by magnetic nanocomposites is not new and has been studied since many years and to my knowledge has never been used at a large scale showing that nothing at this time is able for the same price to replace the typical adsorbents. However, the second part of the paper (modeling) is an important work so the paper can be published but after major corrections. From a general point of view, most of the characterizations are useless and particularly IR and ATG taking into account of the aim of the paper. Here are the comments in more details...

p1 l22 : 59 m2/g is not relatively high for an absorbent but relatively low...

p2 l46-47 : the cost to produce CNTs is certainly higher than to produce activated carbon...

From a general point of view the introduction is well written, but the bibliography not adapted at all and particularly from ref 12 to 24....  Many authors have used CNTs and iron oxides as adsorbents before  2017 and 2018...  Many improvements and more serious bibliography in this area have to be made.

p2 l 88 : I don't understand the sentence : how can materials can lose their mass ????

p3 l102  : it should be better to show a picture of the NTCs (length, diameter, opened or not...).

p 3 l129 : if we look at the TGA curves, if vacuum is used for outgassing as usually, I am not sure that the alginates are so stable during 3h00 at 250°C . The authors have to do the experiment and to weight the sample before and after...

p5 : in the international system angstroems are not used, but nanometers

there a big problem with fig 1 and table 1 : if we look at the figure 1  only CNTs have microporosity and they are the materials which possess the higher average pore width ????   Can the authors explains such a value and particularly with CNTs? TEM or SEM images would be welcome.

in table 1 : to present a average pore witdh of a material which has no porosity is a little dangerous... Be careful with the model or justify please.

p6 figure 2 b : I am very suprised that the 002 reflection of the carbon has disappeared in the final composite

As already written, IR and TGA experiments are usefull in this study. On an other hand TGA under vaccum should be interesting to study the stability of the alginates and of the composites at 250°. A TGA realized under air is also necessary to estimate the real amounts of CNTs and iron oxides in the composite.

p7 l277 : "above 845°C only the metal oxides remained" !!!!   Can the authors justify how the CNTS have disappeared in this temperature range under nitrogen!

there is a big problem between curve a and e : more materials remain in the case of alginates in comparison with the composites made of alginates, CNTs and iron oxide...

To do a TGA curve with iron oxide has no sense...

Author Response

Dear Ms. Allen Dou

Assistant Editor,

Applied Sciences

Thanks for your letter and for sending us the referee comments on the manuscript “Effective adsorption of Methylene Blue dye onto magnetic nanocomposites. Modeling and reuse studies” by S. Álvarez-Torrellas, M. Boutahala, N. Boukhalfa and M. Munoz (applsci-609030) that we sent to be considered for publication in Applied Sciences. Please, enclose you can find the revised version of our manuscript.

We are pleased to find that the referees highlighted the suitability of paper for publication after some revision:

Referee 1. “The preparation of the adsorbents is the part that could impart new knowledge in the field. I recommend the acceptance after addressing minor revision”…

Referee 2. “However, the second part of the paper (modeling) is an important work so the paper can be published”…

Referee 3. “The results were well presented”….

General remarks:

We have changed the manuscript according to the suggestions and comments of the referees, which makes the manuscript more interesting and informative. We acknowledge the remarks from the reviewers which helped us to improve the quality and the comprehension of the work.

We have included some new references, in accordance with the suggestions of the referees. Additionally, SEM technique have been used in order to improve the exploring of the morphological properties of carbon nanotubes; and TGA/DTG analysis of the materials under air atmosphere have been carried out.

We submit a copy of the final version of the manuscript, with the changes marked up in red.

The whole manuscript was edited for proper English language, grammar, punctuation, spelling, and overall style by one qualified native English. On the other hand, the references are formatted according to the requirements of Applied Sciences.

We think that these comments support our (already comprehensive) descriptions, as we explain in the appended report.

Yours sincerely,

Prof. Silvia Álvarez-Torrellas and Prof. Moktar Boutahala.

Detailed comments and list of changes:

Effective adsorption of Methylene Blue dye onto magnetic nanocomposites. Modeling and reuse studies” by S. Álvarez-Torrellas et al. (applsci-609030)

Referee 2

Thanks for your comments on the manuscript. We acknowledge the remarks which helped us to improve the quality and the comprehension of the work.

Reviewer comments

The paper "Effective adsorption...." by S. Alvarez-Torrellas is focused in the adsorption of methylene blue onto magnetic nanocomposites. The adsorption of metals, VOCs... in aqueous media by magnetic nanocomposites is not new and has been studied since many years and to my knowledge has never been used at a large scale showing that nothing at this time is able for the same price to replace the typical adsorbents. However, the second part of the paper (modeling) is an important work so the paper can be published but after major corrections. From a general point of view, most of the characterizations are useless and particularly IR and ATG taking into account of the aim of the paper. Here are the comments in more details...

p1 l22 : 59 m2/g is not relatively high for an absorbent but relatively low...

REPLY and MODIFICATION

We are agree with the referee. As it has been discussed before in the response to Reviewer 1, the tested adsorbents did not show high specific surface area values, so the manuscript has been amended in this sense. The Abstract of the manuscript has been amended as follows: “The beads (A/γ-Fe2O3/f-CNT) presented a relatively low BET specific surface area value of 59 m2.g-1.”

The Introduction Section has been amended: “These oxide materials demonstrated a huge potential for their application in wastewater treatment due to their, magnetic properties, relevant selectivity, reactivity and biocompatibility abilities”.

In the Results and Discussion, attending to the results shown in Figure 1 and Table 1, the manuscript has been amended: “Thus, f-CNT material showed a low-moderate specific surface area (265 m2 g-1), since relatively low BET area values were found for γ-Fe2O3 nanoparticles and A/γ-Fe2O3/f-CNT composites, 35 and 59 m2 g-1, respectively”.

Reviewer comments

p2 l46-47 : the cost to produce CNTs is certainly higher than to produce activated carbon...

REPLY and MODIFICATION

We are agree with the referee. The manuscript, Chapter 1. Introduction, has been amended as follows: “Conventional activated carbon materials are commonly used as adsorbents; however, the difficulties for the separation after treatment in some applications limit their use and, in some cases, make the process unfeasible for industrial applications [7-8].

As it has been widely reported in the literature, conventional activated carbons, synthesized from a single precursor, are economical materials used in the adsorbent processes, and even finding specific applications, such as supercapacitors [9] and catalysts [10], due to their relatively low cost”.

“…the use of MWCNTs as nano-adsorbents is limited due to their separation and regeneration difficulties”.

In the Results and Discussion Chapter, 3.8. Adsorbent recycling, the manuscript has been amended as follows: “The results confirmed that A/γ-Fe2O3/f-CNT beads are efficient adsorbents of aromatic compounds from wastewater, since they can be rapidly and effectively regenerated”.

Gomri, F.; Finqueneisel, G.; Zimny, T.; Korili, S.A.; Gil, A.; Boutahala, M. Adsorption of Rhodamine 6G and humic acids on composite bentonite–alginate in single and binary systems. Appl. Water Sci. 2018, 8, https://doi.org/10.1007/s1320. Oladipo, A.A.; Ifebajo, A.O. Highly efficient magnetic chicken bone biochar for removal of tetracycline and fluorescent dye from wastewater: Two-stage adsorber analysis. J. Environ. Manage. 2018, 209, 9-16. Ruiz, V.; Blanco, C.; Santamaría, R.; Ramos-Fernández, J.M.; Martínez-Escandell, M.; Sepúlveda-Escribano, A.; Rodríguez-Reinoso, F. An activated carbon monolith as an electrode material for supercapacitors. Carbon 2009, 47, 195-200. Rodríguez-Reinoso, F. The role of carbon materials in heterogeneous catalysis. Carbon 1998, 36, 159-175.

Reviewer comments

From a general point of view the introduction is well written, but the bibliography not adapted at all and particularly from ref 12 to 24....  Many authors have used CNTs and iron oxides as adsorbents before 2017 and 2018...  Many improvements and more serious bibliography in this area have to be made.

REPLY and MODIFICATION

We thank the referee for this suggestion. We are totally agree with the referee. The authors have accomplished an intensive search and improvement of the scientific literature before 2017, reporting fundamentals on the synthesis and application as adsorbents of the several kinds of materials mentioned in the Introduction of the manuscript, e.g., carbon nanotubes; magnetic iron oxides; activated carbon, graphene, carbon nanotubes-composites; calcium alginate encapsulated with magnetite nanoparticles, activated carbon, carbon nanotubes, graphene oxide, titania nanoparticles and clays.

Next, the references that have been changed in the manuscript are the following:

Yu, J.-G.; Zhao, X.-H.; Yang, H.; Chen, X.-H.; Yang, Q.; Yu, L.-Y.; Jiang, J.-H.; Chen, X.-Q. Aqueous adsorption and removal of organic contaminants by carbon nanotubes. Sci. Total Environ. 2014, 482-483, 241-251. Díaz, E.; Ordóñez, S.; Vega, A. Adsorption of volatile organic compounds onto carbon nanotubes, carbon nanofibers, and high-surface-area graphites. J. Colloid Interf. Sci. 2007, 305, 7-16. Travlou, N.A.; Kyzas, G.Z.; Lazaridis, N.K.; Deliyanni, E.A. Functionalization of Graphite Oxide with Magnetic Chitosan for the Preparation of a Nanocomposite Dye Adsorbent. Langmuir 2013, 29, 1657-1668. Hu, J.; Shao, D.; Chen, C.; Sheng, G.; Li, J.; Wang, X.; Nagatsu, M. Plasma-Induced Grafting of Cyclodextrin onto Multiwall Carbon Nanotube/Iron Oxides for Adsorbent Application. J. Phys. Chem. B 2010, 114, 6779-6785. Gupta, V.K.; Saleh, T.A. Sorption of pollutants by porous carbon, carbon nanotubes and fullerene-An overview. Environ. Sci. Pollut. Res. 2013, 20, 2828-2843. Kokate, M.; Garadkar, K.; Gole, A. One pot synthesis of magnetite-silica nanocomposites: applications as tags, entrapment matrix and in water purification. J. Mater. Chem. A 2013, 1, 2022–2029. Gong, J.-L.; Wang, B.; Zeng, G.-M.; Yang, C.-P.; Niu, C.-G.; Niu, Q.-Y.; Zhou, W.-J.; Liang, J. Removal of cationic dyes from aqueous solution using magnetic multi-wall carbon nanotube nanocomposite as adsorbent. J. Hazard. Mater. 2009, 164, 1517-1522. Fu, Y.; Wang, J.; Liu, Q.; Zeng, H. Water-dispersible magnetic nanoparticle-graphene oxide composites for selenium removal. Carbon 2014, 77, 710-721. Hassan, A.F.; Abdel-Mohsen, A.M.; Fouda, M.M.G. Comparative study of calcium alginate, activated carbon, and their composite beads on methylene blue adsorption. Carbohyd. Polym. 2014, 102, 192–198. Gao, H.; Zhao, S.; Cheng, X.; Wang, X.; Zheng, L. Removal of anionic azo dyes from aqueous solution using magnetic polymer multi-wall carbon nanotube nanocomposite as adsorbent. Chem. Eng. J. 2013, 223, 84–90. Algothmi, W.M.; Bandaru, N.M.; Yu, Y.; Shapter, J.G.; Ellis, A.V. Alginate-graphene oxide hybrid gel beads: An efficient copper adsorbent material. J. Colloid Interf. Sci. 2013, 397, 32–38.

Reviewer comments

p2 l 88 : I don't understand the sentence : how can materials can lose their mass ????

REPLY and MODIFICATION

We thank the referee for this comment. In order to improve the comprehension of the text, the manuscript has been amended as follows: “It is known that a certain amount of the materials used in the adsorption processes could be lost when they are used in powder form after unit operations (e.g., filtration, centrifugation), generating another environmental problem, huge quantities of waste”.

Reviewer comments

p3 l102  : it should be better to show a picture of the NTCs (length, diameter, opened or not...).

REPLY and MODIFICATION

We thank the referee for this comment. The manuscript has been amended: “A schematic figure of the tested multiwalled carbon nanotubes is depicted in the Supplementary Material (Scheme S1). The provided dimensions in the Scheme have been estimated by TEM analysis, considering opened carbon nanotubes”.

Scheme S1. Schematic figure of the commercial multiwalled carbon nanotubes (CNT).

Reviewer comments

p 3 l129 : if we look at the TGA curves, if vacuum is used for outgassing as usually, I am not sure that the alginates are so stable during 3h00 at 250°C . The authors have to do the experiment and to weight the sample before and after...

REPLY and MODIFICATION

We thank the referee for this comment. The authors have repeated the TGA experiment, weighting the sample before and after analysis. Thus, at initial time, the weight of alginate beads was of 7.24600 mg. The thermal analysis results indicated that at a temperature of 250.0098 °C, the weight of the beads was of 6.31851 mg. This represents a weight loss percentage of 12.8%. Then, at a temperature of 550.0191 °C, the weight decayed up to 3.29826 mg. At the end of the treatment, at 894.0914 °C, the weight of the sample was of 1.06702 mg. According to this, the total weight loss was of 6.17898 mg, which represents a total weight loss of 85.2%.

Reviewer comments

p5 : in the international system angstroems are not used, but nanometers

REPLY and MODIFICATION

We thank the referee for this comment. We are agree with the referee. Angstrom units are substituted by nanometers along the whole manuscript.

Reviewer comments

there a big problem with fig 1 and table 1 : if we look at the figure 1  only CNTs have microporosity and they are the materials which possess the higher average pore width ????   Can the authors explains such a value and particularly with CNTs? TEM or SEM images would be welcome.

REPLY and MODIFICATION

We thank the referee for this comment. Carbon nanotubes are materials that show a mainly mesoporous structure; with a contribution of microporosity (~40 cm3 g-1 at low P/P0 values) and an important quantity of meso-macropores (type IVa isotherm, according IUPAC classification). These adsorbents show that both brands of N2 adsorption isotherm are parallel and near to the verticality, characteristic of materials with cylindrical pores, where capillary condensation occurs [1]. The manuscript has been amended in this sense. So it is expectable that these kind of solids present a wider pore size, in comparison to the other materials without a high quantity of transport pores in their structure (γ-Fe2O3 particles and A/γ-Fe2O3/f-CNT beads).

Therefore, the authors have checked that the provided average pore width values in the manuscript are incorrect. The pore diameter of MWCNT was found of 2.5-3.5 nm, estimated following Barrett-Joyner-Halenda (BJH) method [2]. In addition, the pore diameter values of γ-Fe2O3 and A/γ-Fe2O3/f-CNT composites was of 1-1.6 and 1-1.5 nm, respectively. These values were calculated according MP method, derived from DR equation [3].

The corrected pore diameter values are included in the amended manuscript (Table 1).

Table 1. Textural parameters of f-CNT, γ-Fe2O3 and A/γ-Fe2O3/f-CNT beads.

SBET

(m2 g-1)

Sext

(m2 g-1)

VTotal (cm3 g-1)

Pore diameter

(nm)

f-CNT

265

331

1.35

2.5-3.5a

γ-Fe2O3

35

31

0.06

1.0-1.6b

A/γ-Fe2O3/f-CNT

59

103

0.14

1.0-1.5b

aEstimated by BJH model.

bEstimated by MP method.

SEM micrograph of f-CNT material has been provided in Figure 4b. In addition, TEM micrographs of the tested carbon nanotubes have obtained and have been reported in the Supplementary Material (Figure S1). The manuscript has been amended: “Additionally, the morphology of functionalized carbon nanotubes (f-CNT) was explored by transmission electronic microscopy (TEM) in a microscope JEOL JEM 2100 (200 kV, 0.25 nm of resolution)”. “A TEM micrograph of f-CNT material is shown in Figure S1 (Supplementary Material)”. “For carbon nanotubes, it could be appreciated the usual cylindrical shape of f-CNTs with a diameter from 20 to 50 nm and a length of 50-100 nm (Figure 4b, Figure S1)”.

Figure S1. TEM micrograph of f-CNT material.

Zhang, S.; Shao, T.; Bekaroglu, S.S.K.; Karanfil, T. The Impacts of Aggregation of Surface Chemistry of Carbon Nanotubes on the Adsorption of Synthetic Organic Compounds. Sci. Technol. 2009, 43, 5719–5725. Villarroel-Rocha, J.; Barrera, D.; Sapag, K. Introducing a self-consistent test and the corresponding modification in the Barrett, Joyner and Halenda method for pore-size determination. Mesopor. Mat. 2014, 200, 68-78. Sun, J. Pore size distribution model derived from a modified DR equation and simulated pore filling for nitrogen adsorption at 77 K. Carbon 2002, 40, 1051-1062.

Reviewer comments

in table 1 : to present a average pore witdh of a material which has no porosity is a little dangerous... Be careful with the model or justify please.

REPLY and MODIFICATION

We thank the referee for this comment. We are agree with the referee. This question has been discussed in the previous referee suggestion and amended in the manuscript (changes in Table 1 and 3.1. Characterization of the tested adsorbents, all marked in red).

Reviewer comments

p6 figure 2 b : I am very suprised that the 002 reflection of the carbon has disappeared in the final composite.

REPLY and MODIFICATION

We thank the referee for this comment. We are agree with the referee. The manuscript has been amended as follows: “The characteristic peaks of f-CNT were not identified in the patterns of composite material (A/γ-Fe2O3/f-CNT) due to the low initial f-CNT weight ratio used in the preparation of alginate beads (0.4%). In addition, the presence of the peak at 2θ = 43.83o (400) in maghemite XRD pattern caused the disappearance of the characteristic peak of f-CNT at 2θ = 42.81° (100). These findings may be also attributed by the presence of alginate in the material, which reduces the purity of f-CNT in the final composite”.

Reviewer comments

As already written, IR and TGA experiments are usefull in this study. On an other hand TGA under vaccum should be interesting to study the stability of the alginates and of the composites at 250°. A TGA realized under air is also necessary to estimate the real amounts of CNTs and iron oxides in the composite.

REPLY and MODIFICATION

We thank the referee for this comment. We are agree with the referee. TGA/DTG experiments under air atmosphere of γ-Fe2O3 particles, f-CNT, alginate/γ-Fe2O3 beads and A/γ-Fe2O3/f-CNT composite were carried out. The TGA/DTG profiles can be found in the Supplementary Material (Figures S2a-d). The manuscript has been amended: “Thus, TGA/DTG analysis in air atmosphere (from 30 to 900 °C, 10 °C min-1) of γ-Fe2O3 particles, f-CNT, alginate/γ-Fe2O3 beads and A/γ-Fe2O3/f-CNT composite were accomplished”. “TGA/DTG profiles of γ-Fe2O3 particles, f-CNT, alginate/γ-Fe2O3 beads and A/γ-Fe2O3/f-CNT under air atmosphere were obtained (Figures S2a-d in Supplementary Material). For γ-Fe2O3, similar TG profile was observed (Fig. S2a), with weight losses at the same temperatures that those detected under inert atmosphere. A great change was obtained for the TGA/DTG profile of f-CNT (Fig. S2b), observing that practically the total mass of sample was burned off at 900 °C under air. The thermogravimetric profile of A/ γ-Fe2O3 beads in oxidizing atmosphere (Fig. S2c) was similar to that obtained under inert conditions, with a more intense decay in the weight sample at approximately 200 °C. Finally, TGA/DTG curves of A/γ-Fe2O3/f-CNT composite (Fig. S2d) showed a dramatically decreasing in the weight of sample at ~250 °C, remaining a weight content of 7.37% at the final temperature, attributed to the iron oxides amount present in the composite”.

Reviewer comments

p7 l277 : "above 845°C only the metal oxides remained" !!!!   Can the authors justify how the CNTS have disappeared in this temperature range under nitrogen!

REPLY and MODIFICATION

We thank the referee for this comment. We are agree with the referee. As it was commented above, under nitrogen atmosphere, mass of f-CNT sample did not disappear at temperatures higher than 845 ºC. We have amended the manuscript: “Above 845 °C, a γ-Fe2O3 mass weight of 63.22% remained”.

Figure S2. TGA/DTG profiles of (a) γ-Fe2O3 particles, (b) f-CNT, (c) A/γ-Fe2O3, (d) A/γ-Fe2O3/f-CNT beads.

Reviewer comments

there is a big problem between curve a and e : more materials remain in the case of alginates in comparison with the composites made of alginates, CNTs and iron oxide...

REPLY and MODIFICATION

We thank the referee for this comment. We are agree with the referee. The authors have repeated TGA/DTG analysis, obtaining the same results. We attribute the results to the high percentage of organic matter in the composite coming from the alginate, the citrate incorporated to maghemite nanoparticles and the carboxylic acid incorporated to the f-CNT.

Reviewer comments

To do a TGA curve with iron oxide has no sense.

REPLY and MODIFICATION

We thank the referee for this comment. In this case, we have accomplished TGA/DTG studies of the iron oxide (γ-Fe2O3) under air and inert atmosphere in order to compare its thermal degradation to that observed for A/γ-Fe2O3/f-CNT composite.

Many studies on TGA/DTG analysis of iron oxides have been found in the literature. For example, Iyengar et al. [1] studied the synthesis and characterization of magnetite/maghemite core-shell nanostructures and reported the thermal degradation of the material under nitrogen atmosphere. Similar findings regarding to TGA/DTG analysis could be found in the work of Vargas et al. [2], where the thermal decomposition in air atmosphere of γ-Fe2O3 was studied, obtaining a weight loss of ~25%.

Iyengar, S.J.; Joy, M.; Ghosh, C.K.; Dey, S.; Kotnala, R.K.; Ghosh, S. Magnetic, X-ray and Mösbauer studies on Magnetite/Maghemite Core-Shell Nanostructures Fabricated through Aqueous Route. RSC Adv. 2014, DOI: 10.1039/C4RA11283K. Vargas, O.; Caballero, A.; Morales, J. Enhanced Electrochemical Performance of Maghemite/Graphene Nanosheets Composite as Electrode in Half and Full Li-Ion Cells. Acta 2014, 130, 551-558.

Dr. Silvia Alvarez-Torrellas

Assistant Professor in Chemical Engineering

Complutense University

Department of Chemical Engineering and Materials. Fac. C. C. Químicas.

Avda. Complutense s/n, 28040 Madrid-Spain, Tel: +34 91 3944118

Dr. Moktar Boutahala

Université Ferhat Abbas

Laboratoire de Génie des Procédés Chimiques (L.G.P.C.)

Département de Génie des Procédés, Faculté de Technologie

Sétif 19000, Algeria

Reviewer 3 Report

This manuscript presented data sabout the adsorption of methylene blue onto a new prepared adsorbents. 

The results were well presented, however i felt that the manuscript lacked the role of a key variable in these adsorption processes, that is, the influence of the stirring speed on soluter uptake onto the adsorbent. In many adsorption processes this uptake was dependent on the stirring speed (and how the mixture solution-adsorbent was done), because with the correct stirring speed a minimum in the thickness of the aqueous layer was obtained and the uptake maximized. Did the authors consider the above?, if didn´t, why?.

In equation 7 and 8, a strange and unreadable sign appeared, at least in my computer view.

The manuscript is not yet accepted.    

Author Response

Dear Ms. Allen Dou

Assistant Editor,

Applied Sciences

Thanks for your letter and for sending us the referee comments on the manuscript “Effective adsorption of Methylene Blue dye onto magnetic nanocomposites. Modeling and reuse studies” by S. Álvarez-Torrellas, M. Boutahala, N. Boukhalfa and M. Munoz (applsci-609030) that we sent to be considered for publication in Applied Sciences. Please, enclose you can find the revised version of our manuscript.

We are pleased to find that the referees highlighted the suitability of paper for publication after some revision:

Referee 1. “The preparation of the adsorbents is the part that could impart new knowledge in the field. I recommend the acceptance after addressing minor revision”…

Referee 2. “However, the second part of the paper (modeling) is an important work so the paper can be published”…

Referee 3. “The results were well presented”….

General remarks:

We have changed the manuscript according to the suggestions and comments of the referees, which makes the manuscript more interesting and informative. We acknowledge the remarks from the reviewers which helped us to improve the quality and the comprehension of the work.

We have included some new references, in accordance with the suggestions of the referees. Additionally, SEM technique have been used in order to improve the exploring of the morphological properties of carbon nanotubes; and TGA/DTG analysis of the materials under air atmosphere have been carried out.

We submit a copy of the final version of the manuscript, with the changes marked up in red.

The whole manuscript was edited for proper English language, grammar, punctuation, spelling, and overall style by one qualified native English. On the other hand, the references are formatted according to the requirements of Applied Sciences.

We think that these comments support our (already comprehensive) descriptions, as we explain in the appended report.

Yours sincerely,

Prof. Silvia Álvarez-Torrellas and Prof. Moktar Boutahala.

Detailed comments and list of changes:

Effective adsorption of Methylene Blue dye onto magnetic nanocomposites. Modeling and reuse studies” by S. Álvarez-Torrellas et al. (applsci-609030)

Referee 3

Thanks for your comments on the manuscript. We acknowledge the remarks which helped us to improve the quality and the comprehension of the work.

Reviewer comments

This manuscript presented data about the adsorption of methylene blue onto a new prepared adsorbents.

The results were well presented, however i felt that the manuscript lacked the role of a key variable in these adsorption processes, that is, the influence of the stirring speed on soluter uptake onto the adsorbent. In many adsorption processes this uptake was dependent on the stirring speed (and how the mixture solution-adsorbent was done), because with the correct stirring speed a minimum in the thickness of the aqueous layer was obtained and the uptake maximized. Did the authors consider the above?, if didn´t, why?.

REPLY and MODIFICATION

We thank the referee for this comment. The authors have been previously studied the effect of the stirring speed on the adsorption of many pollutants onto alginate beads and their composites; we have found that when the stirring rate was slow, higher was the adsorption uptake, especially in the case of dyes. In this work, the effect of the stirring rate on the adsorption capacity of Methylene Blue was accomplished, finding that under low stirring, the A/γ-Fe2O3/f-CNT composite beads (using an adsorbent dose of 1 g L-1) retained 86.73 mg of dye per gram of adsorbent, while without stirring we found an adsorption capacity of 90.44 mg g-1, with the same adsorbent dose. According to these results, we decided not use stirring in all the accomplished adsorption experiments.

For the mixture adsorbent-dye solution, firstly, Methylene Blue solutions at the required concentration were prepared and then the adsorbent dose was added. After the equilibrium time was reached (established within 24 hours), several samples were taken from the system, centrifuged and further analyzed by using a UV-Vis spectrophotometer. Due to its high solubility in ultrapure water, no precipitation of Methylene Blue in solution was observed.

Same procedure has been carried out by the authors in previous publications [27, 33, 61].

The manuscript has been amended: “The dispersions were maintained at a constant temperature of 25°C, at a natural pH of 5.2. All the adsorption experiments were carried out without stirring”.

Djebri, N; Boukhalfa, N.; Boutahala, M.; Hauchard, D.; Chelali, N.E.; Kahoul, A. Calcium alginate-organobentonite-activated carbon composite beads as a highly effective adsorbent for bisphenol A and 2,4,5-trichlorophenol: kinetics and equilibrium studies. Water Treat. 2017, 83, 294-305. Djebri, N.; Boutahala, M.; Chelali, N.E.; Boukhalfa, N.; Zeroual, L. Enhanced removal of cationic dye by calcium alginate/organobentonite beads: Modeling, kinetics, equilibriums, thermodynamic and reusability studies. J. Biol. Macromol. 2016, 92, 1277-1287. Boukhalfa, N.; Boutahala, M.; Djebri, N.; Idris, A. Maghemite/alginate/functionalized multiwalled carbon nanotubes beads for methylene blue removal: Adsorption and desorption studies. Mol. Liq. 2019, 275, 431-440.

Reviewer comments

In equation 7 and 8, a strange and unreadable sign appeared, at least in my computer view.

REPLY and MODIFICATION

We thank the referee for this comment. We have corrected Equations 7 and 8:

                                                                                                            (7)

                                                                                                                  (8)

qe (mg g-1) is the equilibrium adsorption capacity;

Ce (mg L-1) is the equilibrium MB concentration in solution;

qm (mg g-1) is the maximum monolayer capacity and KL (L mg-1) is the Langmuir constant, indicative of the affinity of the adsorbent towards the pollutant.

KF (L g-1) and 1/n are parameters of Freundlich equation; the value of 1/n is representative of the adsorption intensity.

Dr. Silvia Alvarez-Torrellas

Assistant Professor in Chemical Engineering

Complutense University

Department of Chemical Engineering and Materials. Fac. C. C. Químicas.

Avda. Complutense s/n, 28040 Madrid-Spain, Tel: +34 91 3944118

Dr. Moktar Boutahala

Université Ferhat Abbas

Laboratoire de Génie des Procédés Chimiques (L.G.P.C.)

Département de Génie des Procédés, Faculté de Technologie

Sétif 19000, Algeria

Round 2

Reviewer 2 Report

Remarks have been generally taken into account... so the paper can be published in the present form...